# communications
# earth & environment

# Recent decreases in snow water storage in western North America

Katherine E. Hale[1,2,3✉], Keith S. Jennings [4], Keith N. Musselman [1,2], Ben Livneh [5,6] & Noah P. Molotch [1,2,7]

Mountain snowpacks act as natural water towers, storing winter precipitation until summer months when downstream water demand is greatest. We introduce a Snow Storage Index (SSI), representing the temporal phase difference between daily precipitation and surface water inputs—sum of rainfall and snowmelt into terrestrial systems—weighted by relative magnitudes. Different from snow water equivalent or snow fraction, the SSI represents the degree to which the snowpack delays the timing and magnitude of surface water inputs relative to precipitation, a fundamental component of how snow water storage influences the hydrologic cycle. In western North America, annual SSI has decreased ($p < 0.05$) from 1950–2013 in over 25% of mountainous areas, as a result of substantially earlier snowmelt and rainfall in spring months, with additional declines in winter precipitation. The SSI and associated trends offer a new perspective on hydrologic sensitivity to climate change which have broad implications for water resources and ecosystems.

[1] Department of Geography, University of Colorado at Boulder, Boulder, CO, USA. [2] Institute of Arctic and Alpine Research, University of Colorado at Boulder, Boulder, CO, USA. [3] Department of Civil and Environmental Engineering, University of Vermont, Burlington, VT, USA. [4] Lynker, Boulder, CO, USA. [5] Cooperative Institute for Research in Environmental Sciences, University of Colorado at Boulder, Boulder, CO, USA. [6] Department of Civil, Environmental, and Architectural Engineering, University of Colorado at Boulder, Boulder, CO, USA. [7] Jet Propulsion Laboratory, California Institute of Technology, Pasadena, CA, USA. ✉email: Katherine.E.Hale@colorado.edu

In western North America, the storage of cold-season precipitation in mountain snowpacks, and subsequent snowmelt in spring and early summer months, sustain streamflow and provide water for downstream needs when atmospheric, ecological, and societal demands are greatest[1–7]. In the last century, climate warming has been linked to precipitation phase shifts from snowfall to rainfall[8] decreases in peak snow water equivalent (SWE)[6] earlier snowmelt onset[9–11], and shorter snow cover duration[12,13]. These changes have likely reduced the ability of mountain snowpacks to store water as snow[14]. A shift in timing between cold-season precipitation and subsequent snowmelt, and the relative magnitudes, represents an under-studied yet critical dimension of climate change impacts on water resources. Concerningly, changes in the amount and timing of water release from mountain snowpacks have cascading effects on streamflow timing and volumes[11] and water storage and conveyance infrastructure, and therefore on regional water availability[11,15–19].

SWE and snowfall fraction (i.e., the percentage of annual precipitation falling as snow) have been used as proxies for annual water delivery as streamflow[6,20–23]. However, these variables do not describe the essential role that snowpack water storage plays in creating a temporal lag between precipitation inputs and its availability to watersheds to eventually become streamflow, evapotranspiration, or soil/ground water. Across regions where peak or April 1 SWE and annual snowfall fraction or snow cover duration may look similar, snow water storage can vastly differ, indicating potentially large differences in hydrologic sensitivity to climate change. Complex regional variability in snowpack sensitivity to climate change is inter-linked with how water resources are partitioned among evapotranspiration, streamflow, and soil water storage, as protracted snow cover duration aligns water availability with atmospheric demand[20,24]. The use of a single metric to assess trends in both the temporal differences and relative magnitudes of precipitation and corresponding surface water inputs, as rainfall and snowmelt, across regions has critical implications to better monitor ecosystem stress and inform water resource management[25–27].

In this study, we develop a new metric, widely applicable for characterizing the magnitudes and temporal offset between precipitation and surface water inputs, which quantifies the ability of mountain snowpacks to act as a natural reservoir. Surface water inputs, in the context of this work, represent the per grid cell input of water to the terrestrial system from the atmosphere via rainfall or snowmelt only, and thus do not include lateral movement of soil water, ground water, or streamflow from one grid cell to another. This perspective on snowfall, snowmelt (magnitude and timing), and seasonal snow water storage targets an unaddressed gap in hydrology by evaluating how future changes in climate may modify local and regional water availability through changes in the timing of water inputs to the terrestrial system relative to the timing of precipitation. We ask: *How does the timing and magnitude of snow water storage vary across western North America and how has it changed in recent decades (i.e., 1950–2013)?*

## Results

**Long-term Snow Storage Index**. We calculated long-term (1950–2013) and annual Snow Storage Index (SSI) values across western North America to assess snow water storage. The storage of water in the snowpack creates a temporal offset between precipitation and later surface water inputs. The SSI is thus a numeric comparison of the phase and amplitude of sine curves fit to average daily precipitation and modeled average daily surface water inputs (see "Methods" section). An SSI of −1 indicates that precipitation and surface water inputs are highly seasonal and in phase with one another, an SSI of 0 indicates no seasonality in precipitation or surface water inputs, and an SSI of 1 indicates precipitation and surface water inputs are highly seasonal and out of phase with one another. We therefore expect mountain regions with relatively substantial and persistent snowpacks to have positive SSI values (≥0), as their snowpacks act as natural water towers, mediating a delay and causing a phase shift between precipitation and surface water inputs.

To calculate the SSI across western North America, we used precipitation forcings and SWE outputs from the Variable Infiltration Capacity (VIC) model[28,29] (see "Methods" section). Fifty percent of the western North American domain had long-term SSI values greater than 0 (Fig. 1a). Large areas of high SSI values (>0.75) were most prominent in mountainous regions, particularly in the ecoregions of the North Cascades, Cascades, Columbia Mountains/Northern Rockies, Canadian Rockies, Sierra Nevada and the Middle Rockies (Fig. 1a). Moderate SSI values (0.5–0.75) existed in the Middle Rockies, Wasatch/Uintas, and Southern Rockies, with low SSI values (0–0.5) existing around the perimeter of most ecoregions (see long-term average SSI histograms per ecoregion in Supplementary Fig. 1).

As an illustration of the SSI concept, consider a grid cell in the central Sierra Nevada, which is heavily dominated by late fall and winter snowfall, has a high long-term SSI value (0.94), as there is an approximate 6-month lag between the timing of precipitation (i.e., snowfall) and surface water inputs (i.e., snowmelt), with distinct seasonality in both variables (Fig. 1b). This example contrasts with a grid cell in the Front Range of Colorado in the Southern Rockies, an area with marked spring precipitation. Here, precipitation and surface water inputs are more in-phase with one another and precipitation lacks seasonality, yielding an SSI value of 0.32 (Fig. 1c). Thus, greater SSI values (>0.75) occur when large amounts of precipitation (as a depth of snowfall) occur in the fall and winter months with large amounts of surface water inputs (particularly as snowmelt) generated several months later. Lower SSI values (0–0.5) occur in areas with lower precipitation seasonality and/or smaller snowpacks, such as in the example of the Colorado Front Range. These two contrasting grid cells within the Sierra Nevada and Southern Rockies ecoregions accumulate peak SWE depths within 20% of each other, snow fractions within 5% of each other, and maximum SWE dates within one week of each other[30]. Yet, these contrasting locations, intended to illustrate the SSI concept, behave considerably different with regard to snow water storage, as represented by SSI. As such, SWE or elevation do not alone serve as proxies for snow water storage, and by grid cell, we found no significant correlations between long-term SSI and SWE or elevation (shown for select ecoregions in Supplementary Fig. 2).

### Spatial trends in the Snow Storage Index
*Modeled data.* Ninety-two percent of the study area with SSI values ≥ 0 exhibited a decrease in SSI (max decline: −0.03/decade, mean: −0.005/decade), and the other 8% showed an increase in SSI (Fig. 2a). 25.1% of the study domain exhibited a statistically significant decline in SSI ($p < 0.05$, max decline: −0.03/decade, mean: −0.01/decade) while only 0.9% showed a significant increase in SSI (Fig. 2b). Decreases in SSI, where SSI ≥ 0, were particularly widespread in the Canadian Rockies (64% of the ecoregion), Columbia Mountains/Northern Rockies (45% of the ecoregion), and in the Idaho Batholith (27% of the ecoregion). Declines in SSI in the Cascades (17% of the ecoregion) and Southern Rockies (15% of the ecoregion) were less widespread.

SSI declines were primarily driven by two climate-related mechanisms: (1) shifts in snowmelt timing and (2) declines in

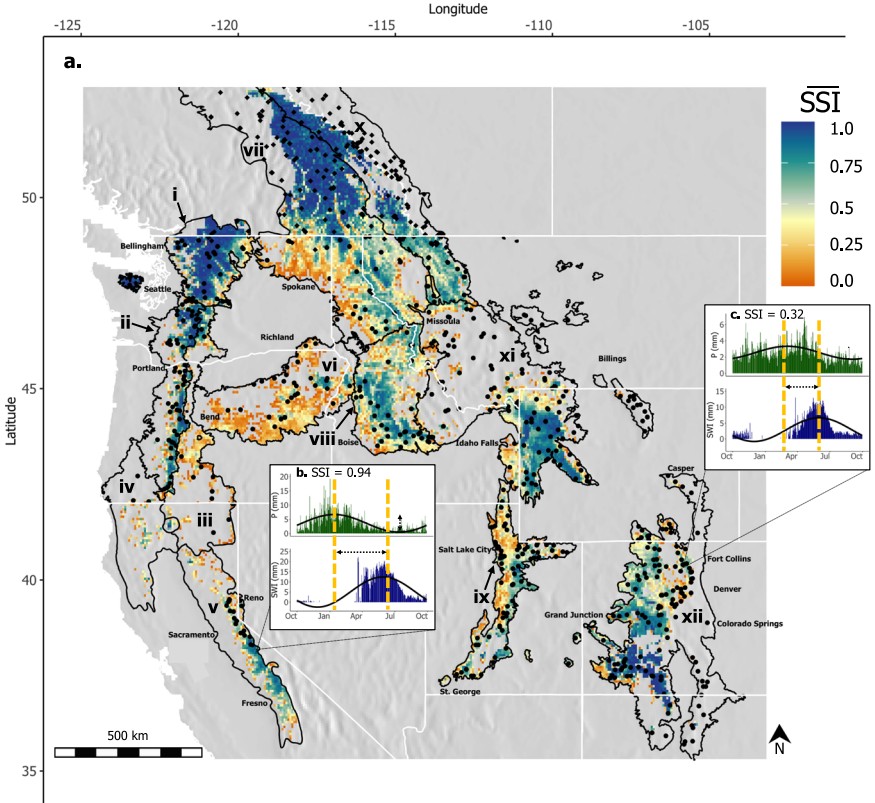

**Fig. 1 Long-term Snow Storage Index (SSI) across Western North America. a** Long-term average SSI across the mountainous western cordillera, constrained to areas within each EPA Level III ecoregion where $\overline{SSI} \geq 0$ for 1950-2013. Black points represent SNOTEL stations in the USA, and black diamonds represent CanSWE stations in Canada. Spatially distributed VIC modeled $\overline{SSI}$ data are shown in color, with darker blue shades indicating greater $\overline{SSI}$ values and darker orange shades lower values. The twelve EPA Level III ecoregions of the western cordillera are outlined in black and are labeled as: (i) North Cascades, (ii) Cascades, (iii) Eastern Cascade Slopes and Foothills, (iv) Klamath Mountains, (v) Sierra Nevada, (vi) Blue Mountains, (vii) Columbia Mountains/Northern Rockies, (viii) Idaho Batholith, (ix) Wasatch and Uinta Mountains, (x) Canadian Rockies, (xi) Middle Rockies, (xii) Southern Rockies. Panels **b** and **c** show exemplar grid cell daily average precipitation (P) and surface water inputs (SWI) data used to calculate $\overline{SSI}$ in the Sierra Nevada, and Southern Rockies, respectively. Sine curves fit to the data are shown with black lines and the vertical yellow dashed lines indicate the peak of each sine curve. The horizontal dashed arrows notate the temporal difference in the peak precipitation and peak surface water inputs. The vertical dotted arrows notate the amplitude of each select curve.

precipitation seasonality. The first mechanism is a result of warming-related shifts in surface water input generation through earlier snowmelt onset and rainfall. For example, the Cascades exhibited a decline in SSI driven by a shift to earlier seasonal surface water inputs (dotted sine curves; Fig. 2b.i). In this respect, peak surface water inputs for the second half of the record (1982–2013) shifted almost a month earlier in time compared to the first half of the record (1950–1981). Thus, the timing of surface water inputs has become more in-phase with the timing of precipitation, causing the SSI to decrease. This phase shift of the surface water input curve, which represents earlier snowmelt or rainfall, has occurred across all ecoregions in the study domain with statistically significant decreases in SSI (not shown per ecoregion). The second mechanism causing declines in SSI is related to seasonal trends in precipitation. For example, in the Canadian Rockies and Columbia Mountains/Northern Rockies, decreases in SSI were attributable, in part, to decreases in winter precipitation. Notable declines in December, January, and February snowfall have reduced the seasonality of precipitation through time and subsequently reduced the amount of surface water inputs generation later in the year (Fig. 2c.ii, solid red line). Small aerial increases in SSI were seen in locations where precipitation seasonality increased due to increases in winter precipitation.

*Observed data.* Observed SSI trends across the study area corroborate the model-based analysis, with 80% of Snowpack Telemetry (SNOTEL) stations recording a decrease in SSI value over the period of 1984–2018 (Fig. 2c, circular points below the black Canadian border). The maximum decline in SNOTEL-derived SSI was −0.06/decade with an average −0.005/decade. Statistically significant trends in observed SSI ($p < 0.05$) existed for 17.8% of all SNOTEL stations, also largely negative in slope (Fig. 2d; max decline: −0.06/decade, mean: −0.01/decade). By ecoregion, the difference in SSI slope between the observed SNOTEL and modeled datasets ranged from 0.001/decade to 0.083/decade. Declines in SSI in both observed and modeled datasets were consistent with related declines in SNOTEL-reported SWE across this region and time period[6,13]. Inter-annual variance in the SSI in both modeled and SNOTEL datasets are shown in Supplementary Fig. 3.

Only April 1 SWE data were available within the observational Canadian historical SWE (CanSWE) dataset[31], with varying record lengths between stations, which are represented as diamonds above the Canadian border in Fig. 2c and d (see "Methods" section). April 1 SWE in this region has predominantly declined through time (Fig. 2c, 66% of all CanSWE stations). Statistically significant trends ($p < 0.05$) in April 1 SWE existed in 18.7% of all CanSWE stations (Fig. 2d, diamonds).

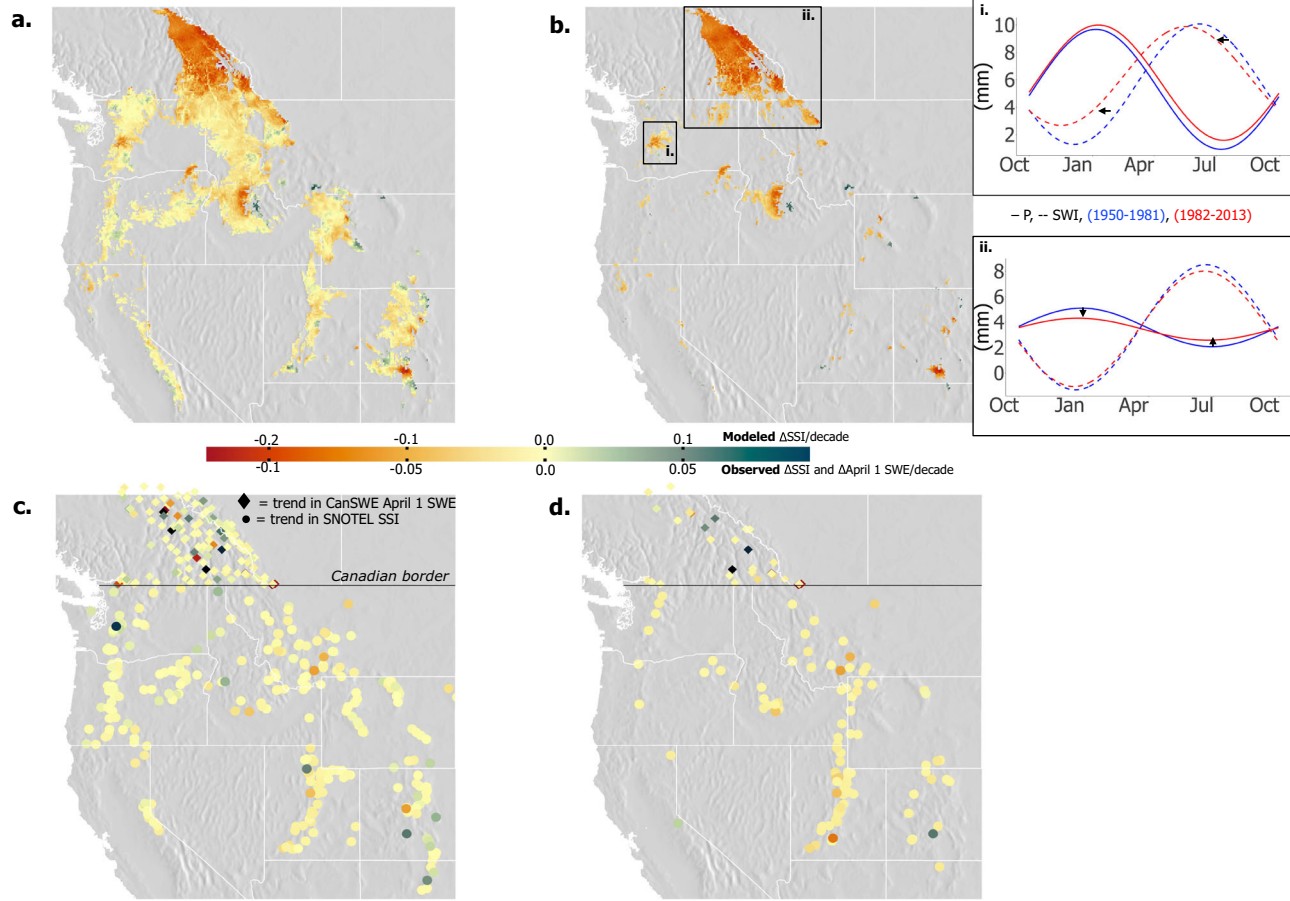

**Fig. 2 Modeled and observed change in SSI from 1950–2013. a** The annual VIC-based change in SSI, reported as ΔSSI per decade (1950–2013), for all grid cells with $\overline{SSI} \geq 0$, and **b** for grid cells where the change in SSI is statistically significant ($p < 0.05$). Panel **c** shows the SNOTEL-based annual ΔSSI per decade (circles, 1984–2018) and CanSWE-based annual ΔSWE for 1 April per decade (diamonds, 1928–2020). The Canadian border is represented with a horizontal black line. Panel **d** shows the areas where changes in SNOTEL-derived SSI and CanSWE 1 April SWE were significant ($p < 0.05$), representing 17.8% ($n = 97$) of SNOTEL stations and 18.7% ($n = 37$) of CanSWE stations. Subpanels in **b** show sine curves fit to long-term, daily average precipitation (solid line) and surface water inputs (SWI, dashed line) for the first and second half of the record (blue: 1950–1981 and red: 1982–2013) in the (**i**) Cascades and (**ii**) Canadian/Northern Rockies.

**Average trends in the Snow Storage Index**. In addition to the grid cell and station-level trends, we were also interested in how SSI values were changing across western North America as a whole. In other words, have the region's mountains been losing their ability to act as natural water towers? We spatially averaged modeled and observed SNOTEL SSI, incorporating a weighted grid cell area to account for slight differences in grid cell size at different latitudes[32]. We found that the average modeled SSI across western North America exhibited a statistically significant decline from 1950–2013 ($p < 0.01$, Fig. 3a). The declining trend was also significant when evaluating the modeled SSI while excluding the Canadian Rockies and Columbia Mountains/Northern Rockies ($p < 0.05$), the two ecoregions that expressed the majority of statistically significant SSI declines (not shown). A decadal analysis of the modeled data revealed similar decreasing SSI trends when averaged across the entire area (Fig. 3b) and by grid cell (Supplementary Fig. 4) compared to the annual analysis. The observational SNOTEL data showed an average decline in SSI without a significant trend (Supplementary Fig. 5), although as noted, the observational record length is 30 years shorter than the modeled record length (34 years vs. 64 years).

Finally, to further the robustness of the SSI trend analysis, we evaluated the centroid and centroid timing of precipitation and surface water inputs (in mm). Most ecoregions showed similar trends in these metrics with that of SSI trends. The areal extent of grid cells experiencing a statistically significant trend in the centroid analysis was within 5–15% of the area that exhibited a significant SSI trend (not shown). However, while the centroid analysis is complementary to the SSI analysis, the magnitude of the centroid is insensitive to the seasonality of precipitation or surface water inputs. The SSI incorporates the seasonality of these variables within the amplitude of the sine curves, in addition to the phase and temporal differences[33,34] (see "Methods").

**Drivers of the Snow Storage Index decline**. Averaging across all grid cells with a statistically significant decline in SSI (Fig. 2b), snowmelt and rainfall increased in March (Fig. 4a) and snowmelt successively decreased in July ($p < 0.05$, Fig. 4b) without a significant change in precipitation magnitude (mechanism shown in Fig. 2b.i). The change in surface water input timing across this area represents a shift toward earlier snowmelt and rainfall and therefore a reduction in the temporal offset between the timing of precipitation and timing of surface water inputs. This offset shift is the basis of the SSI. Further, the warming-induced increase in surface water inputs in March, coupled with a decrease in surface water inputs in July, provides a complementary signal to the declining SSI trend which is independent of the sine curve functions in the SSI calculation (see "Methods"). In some northern

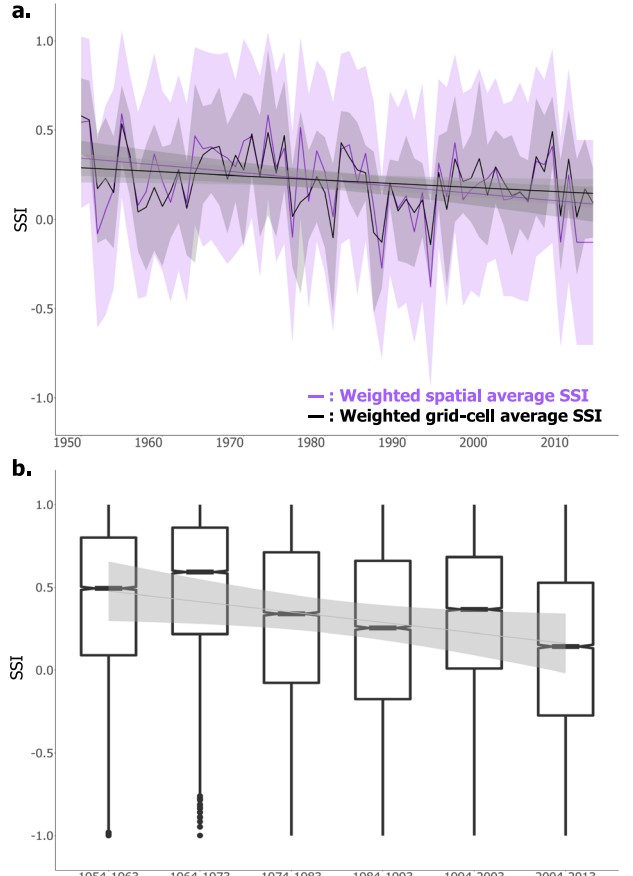

**Fig. 3 Spatial average modeled SSI has significantly declined from 1950-2013. a** Spatially weighted average VIC-based annual SSI with daily precipitation and surface water inputs averaged spatially prior to the SSI calculation (purple line), and with SSI spatially averaged by grid cell after each grid cell SSI was calculated (black line); standard errors are shown with purple and black shading, respectively. Trend lines for SSI through time are shown in both cases ($p < 0.01$). **b** Average modeled decadal SSI across the study region, showing the median, interquartile range, and outliers across the 6 decades over the VIC record. Trend line indicates $p < 0.05$, with standard errors shown with black shading.

ecoregions, predominant decreases in precipitation seasonality, by way of decreased winter precipitation, resulted in decreased amplitudes of the related precipitation sine curves, which reduced SSI (Fig. 2b.ii). Specifically, precipitation in December, January, and February has significantly decreased in the Canadian Rockies and Columbia Mountains/Northern Rockies which, combined with earlier snowmelt, significantly decreased surface water inputs in June and July (Supplementary Fig. 6).

**Climate sensitivities of the Snow Storage Index**. To explore the climate sensitivity of SSI, we conducted a correlation analysis between SSI and commonly studied hydroclimatic variables[6,35]. Correlations were considered significant if $p < 0.05$ and $-0.5 > r > 0.5$. SSI was positively correlated with annual snowfall fraction in all ecoregions (Fig. 5a). SSI was positively correlated with precipitation in fall and winter seasons (Fig. 5b, c) but negatively correlated with precipitation in spring months in intermountain and continental ecoregions (Fig. 5d). Spring precipitation, even when occurring as snowfall, acts to align the timing of precipitation and surface water input generation and thus reduces SSI. Air temperature in maritime and northern ecoregions was consistently negatively correlated with SSI (Fig. 5a–d).

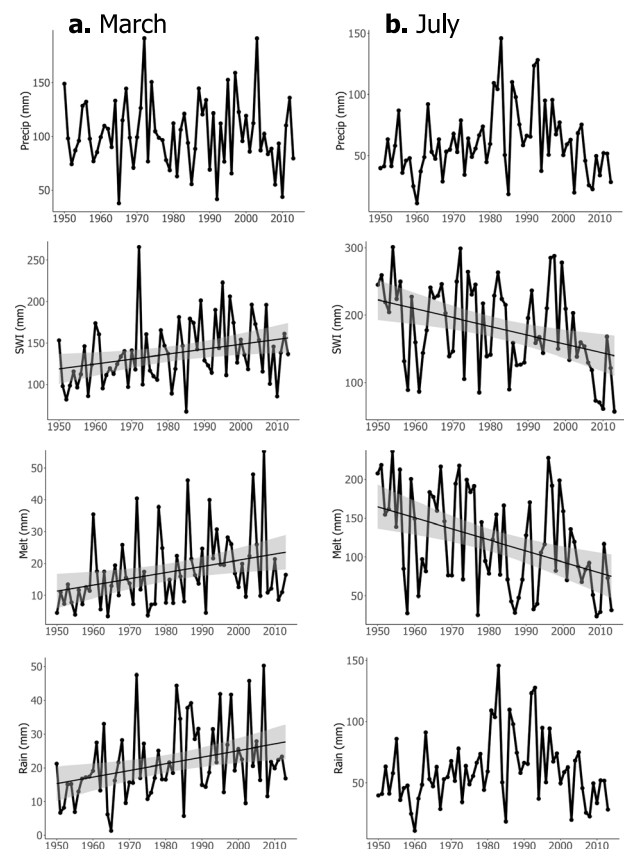

**Fig. 4 Surface water inputs increase in spring and decrease in summer months.** Averaged across grid cells where SSI is significantly declining, spatial average monthly precipitation, surface water inputs (SWI), snowmelt, and rainfall in **a** March and **b** July from 1950–2013; trend lines indicate $p < 0.05$.

Evaluated across the entire study area, areas and years with greater SSI values (>0.75) and thus the largest SSI anomaly percentile bins were associated with cold and wet conditions in fall and winter months but dry conditions in spring months (Fig. 6a±d, blue points). This result further emphasizes the aforementioned distinction between snow-influenced regions that are dominated by fall and winter snowfall versus spring snowfall. Anomalously low SSI values (0–0.5) and thus the smallest SSI anomaly percentile bins were associated with annually dry conditions (Fig. 6a, red points). Further, the lowest SSI values/percentile bins coincided with above-normal air temperatures and with anomalously wet spring (Fig. 6d) and summer (not shown) conditions. Averaged by individual ecoregion, the SSI anomaly percentile bins generally followed the anomalous SSI, precipitation, and temperature patterns seen across the entire study area (Fig. 6a–d, gray lines). Correspondingly, greater SSI values (>0.75) and thus greater SSI anomaly percentile bins occurred in areas and years that received a greater fraction of annual precipitation in fall and winter months (Fig. 6e, f). Conversely, lower SSI values (0–0.5) and thus lower SSI anomaly percentile bins occurred when spring precipitation fraction was high (Fig. 6g). SSI anomaly and precipitation fraction values averaged by individual ecoregions followed these patterns (Fig. 6e–g, gray lines).

**Comparison of modeled SSI to observational datasets**. SNOTEL-derived SSI values exhibited a statistically significant relationship to VIC model SSI values (Supplementary Fig. 7, $p < 0.05$,

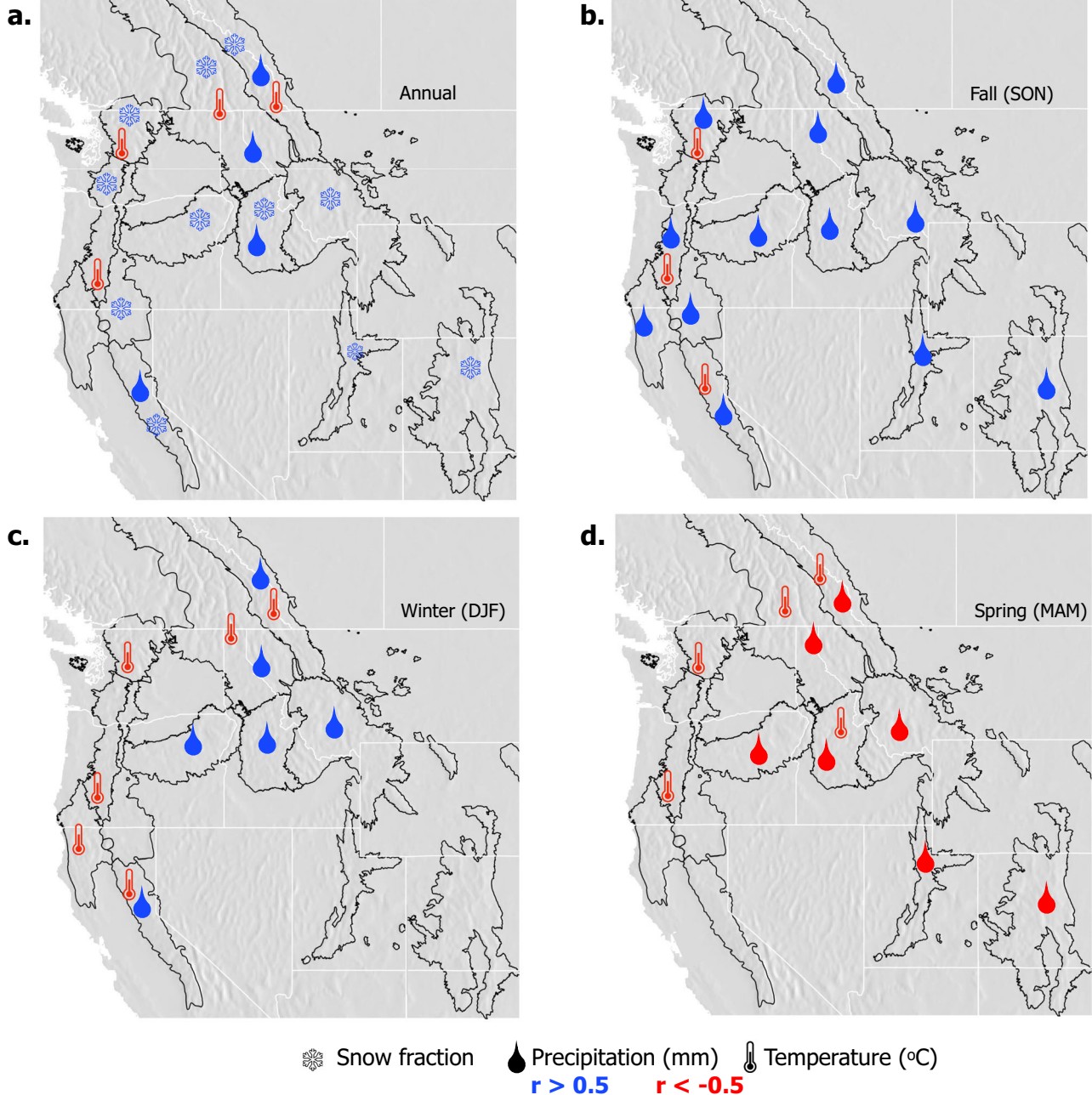

**Fig. 5 SSI was negatively correlated with temperature in all seasons and positively correlated with precipitation, except in spring months.** Correlation analysis determine climate attributions of SSI changes, per ecoregion through the 64-year record. Average temperature, total precipitation, and annual snow fraction were compared to annual $\overline{SSI}$. Variables were averaged spatially across each ecoregion. Hydroclimatic variables significantly correlated ($p \leq 0.05$) with annual $\overline{SSI}$ (1950–2013) for EPA Level III ecoregions for **a** annual average, **b** fall season (September, October, November); **c** winter season (December, January, February), and **d** spring season (March, April, May); red = negative correlation, blue = positive correlation ($r < -0.5$ or $r > 0.5$).

$r^2 = 0.51$), providing confidence that the model output was appropriate to estimate long-term average SSI. Compared to SNOTEL-derived SSI, VIC-derived SSI calculations underestimated larger SSI values and overestimated smaller SSI values, as both the linear and Theil-Sen[36,37] slopes were less than 1. Deviations between modeled and observed SSI may be a result of differences in spatial scale between the VIC model (6 km grid cells) and the SNOTEL stations (point scale). Only April 1 SWE data were available within the CanSWE dataset, so we were unable to generate comparable annual SSI values in the Canadian portion of the modeling domain. However, a comparison between VIC modeled and CanSWE observed

April 1 SWE showed a similar relationship as the VIC-SNOTEL SSI relationship ($p < 0.05$, $r^2 = 0.47$).

## Discussion

Declines in grid cell and region-wide annual SSI values across western North America (Figs. 2a and 3a) are primarily a result of a shift in the timing of surface water inputs toward earlier in the year (Figs. 2b.i and 4a), consistent with previously documented trends in declining snowfall fractions[8,35] and earlier snowmelt[10,38]. In select ecoregions (e.g., Canadian Rockies and Columbia Mountains/Northern Rockies), there exists a secondary

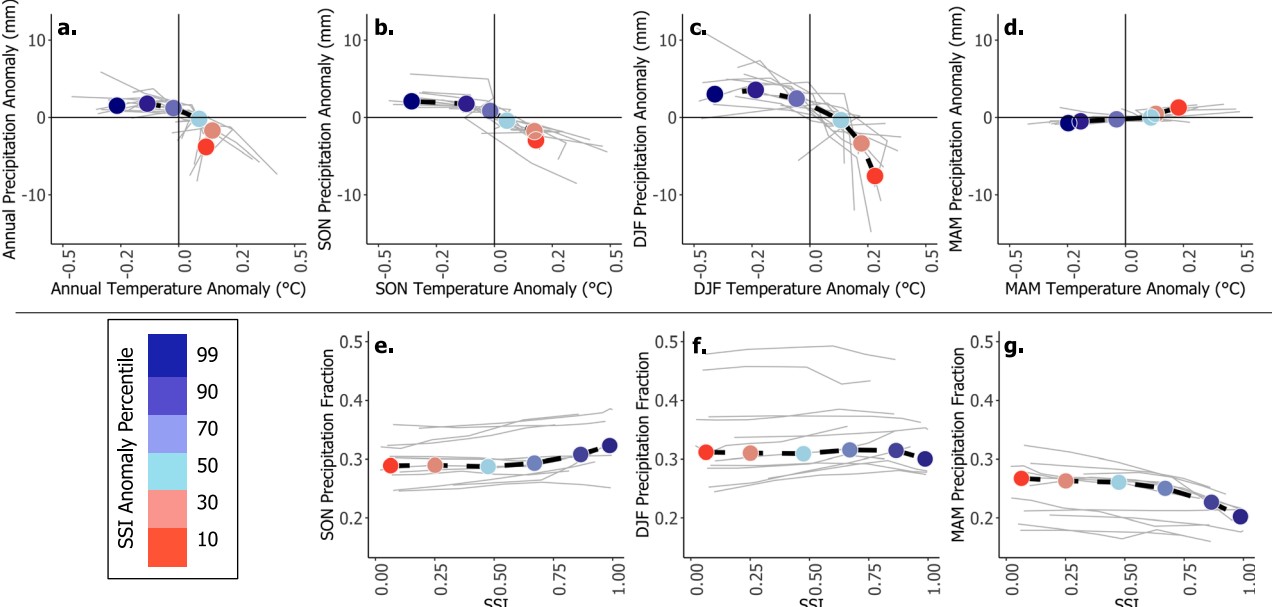

**Fig. 6 Greater SSI values occurred under relatively cool and wet conditions, with low spring precipitation fractions. a–d** VIC-forcing average air temperature anomaly (horizontal axis) versus average precipitation anomaly (vertical axis), averaged by SSI anomaly percentile across the entire study area, and for the fall (SON), winter (DJF), and spring (MAM) seasons, respectively. Panels **e–g** show the average SSI value per SSI anomaly percentile bin (horizontal axis) versus the fraction of annual precipitation occurring within the fall, winter, and spring seasons, respectively, also across the entire study area. All data points are colored by the respective SSI anomaly percentile bin with larger SSI anomalies in blue and lower SSI anomalies in red. Each gray line represents the respective average temperature and precipitation anomaly values, and precipitation fraction and SSI values per individual ecoregion also grouped by SSI anomaly percentile bin.

mechanism for SSI declines, which is a declining trend in winter precipitation and thus annual precipitation seasonality[39] (Supplementary Fig. 6). As the climate continues to warm, further decreases in the SSI will likely occur via continued snowfall fraction declines, earlier snowmelt, and a reduced temporal phase shift between precipitation and surface water inputs[39,40] (Figs. 5a–d and 6e–g). Regions with a greater SSI (>0.75) that are dominated by fall and winter snowfall and thus exhibit a longer temporal delay between precipitation and surface water inputs will potentially see greater hydrologic sensitivity to prolonged warming due to more drastic changes in the temporal phase difference between precipitation and surface water inputs as the climate changes. Susceptibility to secondary climate perturbations (e.g., extreme events, and changes in vegetation) may also be enhanced in these same areas[41,42]. The hydrology of areas experiencing lower SSI values (0–0.5) by way of less seasonal precipitation and/or reduced surface water inputs may exhibit less hydrologic sensitivity to changes in snowpack water storage associated with warming. Thus, trends in SSI represent a new indicator of the hydrologic sensitivity to climate change, signifying the declining ability of western mountains to act as natural water towers, storing water as snow until later in the year.

Importantly, the SSI metric quantifies the temporal offset between precipitation and surface water inputs, in addition to the amount of water stored in the snowpack, a combination of hydrologic characteristics not previously captured in a single value. The SSI complements preceding indices aimed at quantifying the integrative hydrologic impact of seasonal snow, such as the temporal lag between precipitation and snowmelt used to identify downstream hydrologic deficits from snowmelt and rainfall[43] and the potential for snowmelt to meet downstream demands in hydrologically vulnerable regions of the world[4,5]. Ecosystems rely on relatively predictable offsets between precipitation and snowmelt[44,45]. Especially in Mediterranean

climates, which receive little surface water inputs from rainfall throughout the summer months[46], a shift toward earlier surface water inputs (Fig. 4a) may result in water stress on vegetation later in the growing season, during periods of greater atmospheric water demand[45,47,48]. Fall season precipitation shifts from snowfall to rainfall and/or increases in surface water inputs by way of early season melt would similarly decrease SSI. This fall season signal was not explicitly evaluated herein as the climate sensitivities during this period were modest relative to spring and early summer. Winter or early spring rain-on-snow events would also act to align precipitation and surface water input timing and thus decrease the SSI. Hence, SSI is a useful metric of snow water storage that is applicable to hydrological processes, which has not otherwise been fully captured by previous metrics. The SSI and associated sine curve methodology (see "Methods") can also be applied to annual to decadal and long-term timeframes.

Further utility in the SSI is expected when evaluating hydrologic partitioning to streamflow and sensitivities to climate change. Because the increase of water availability in March and decrease in July has been documented in flow volumes across the western United States[11], we would expect the SSI and its components to be highly related to the ultimate amount of water which becomes streamflow for downstream users. Future work should further examine how the decaying of mountain snow water storage manifests in the context of Earth's ecosystems, water cycle, and water security.

**Assumptions and limitations**. The VIC model employs a default rain–snow air temperature range centered around 0.0 °C to partition precipitation into rainfall and snowfall[29]. Future work should explore alternative rain–snow air temperature thresholds to evaluate associated sensitivities of snowfall fraction and snowmelt model outputs[49,50]. In addition, a more exhaustive consideration of snowmelt model uncertainties (e.g., forest-vegetation) is warranted given their influence on the timing and

amount of surface water input generation[51]. Similarly, while the effects of the subsurface are expected to be minimal with regards to surface water inputs, relevant connectivity among soil, ground water, and vegetation should also be examined, particularly as it pertains to eventual streamflow[15,52]. Finally, higher resolution datasets in complex, mountainous terrain could potentially reduce errors in meteorological forcings[53].

## Methods

**Study area.** Western North America was chosen as the study area for the following reasons: (1) seasonal snowpack is the primary source of water resources for agriculture, industry, and other purposes in this region; (2) snowmelt-induced runoff from regions storing water as snow across the western United States accounts for ~70% of its annual runoff[54], where over 60 million people depend on snowmelt for downstream purposes[1]; (3) there are abundant snow, water balance, and meteorology data (e.g., SWE, precipitation, temperature) across the region from in situ observations, remote sensing products, and physically based hydrologic models.

The study area for this work includes the major mountainous Level III ecoregions of western North America as defined by the US Environmental Protection Agency[55] (Fig. 1a), including: North Cascades, Cascades, Eastern Cascades Slopes and Foothills, Sierra Nevada, Columbia Mountains/Northern Rockies, Klamath Mountains, Blue Mountains, Idaho Batholith, Canadian Rockies, Middle Rockies, Wasatch and Uinta Mountains, and Southern Rockies. The average elevation of these ecoregions is 1630 m, ranging from 100 m to 4410 m[19,56].

**Datasets.** We employed a gridded hydrometeorological dataset that contains spatially and temporally continuous daily meteorological forcings and simulated Variable Infiltration Capacity (VIC) model states and fluxes at 1/16° (~6 km) resolution from 1950 to 2013[29,57]. Within the suite of model inputs and outputs (see full list at: https://vic.readthedocs.io/en/master/Documentation/Drivers/Classic/ClassicDriver/), we focused on: station-derived precipitation and simulated SWE. Snowmelt was calculated as the negative change in daily SWE. VIC considers blowing snow sublimation within its snow model but no lateral transport of wind-blown snow across grid cells[58]. In general, with relatively large grid cells, the amount of snow distribution across grid cells is assumed to be negligible relative to the snow fluxes within grid cells[20,59]. We assumed that negative changes in SWE, by way of a latent heat flux, were primarily associated with a melt flux as opposed to sublimation, which may introduce some uncertainty in our analysis[20,60,61]. However, we would not expect small residuals of snow lost to sublimation to affect our analysis, since the SSI trends were consistent across differences in climate and relative humidity, known drivers of snowpack ablation[62]. Finally, surface water inputs were assumed to be the summation of snowmelt and rainfall at each timestep, distinctly different from quantifying surface water or streamflow, as we focused on water availability from the atmosphere at the terrestrial surface and not delivery downstream. Similarly, an evaluation of soil and ground water storage are also outside the scope of this work.

The VIC model has been previously applied to simulate the mountain snowpack in many works[6,40,63–66], showing similarities in environmental conditions and outputs with other land surface models[48,67,68]. Subsequently, the data generated from VIC have provided hydrologic estimates consistent with observations[68]. This study used the data from the VIC version and parameterization of previous works[28,29], which was validated against streamflow observations for the major river basins of the conterminous U.S. This dataset[28,29] was masked to the domain of interest (Fig. 1a). A full iterative energy balance option was used within the previous VIC simulations[28,29], while an explicit frozen soil option was not selected given the additional uncertainties introduced by the complexity of the frozen soil routine and its numerical instabilities. The lack of frozen soils are expected to have negligible impact on SSI, since they primarily affect the fate of meltwater, i.e., infiltration versus overland flow.

For validation of the modeled Snow Storage Index (SSI, see next section) and additional observation-based analysis, ground-based snow observations were used from the automated SNOwpack TELemetry (SNOTEL) network in the western United States[69], and the Canadian historical Snow Water Equivalent dataset (CanSWE) was used in Canada[31]. The SNOTEL network provides continuous snow-pillow SWE, precipitation, and temperature measurements at the point-scale in real-time[70,71]. Precipitation data from each SNOTEL location ($n = 730$) from years 1984–2018 were partitioned into rainfall or snowfall using an approach where a spatially variable rain–snow air temperature threshold was extracted for each station[49]. While the SNOTEL network came online in 1979, most (>75%) of the stations within the study domain began collecting data in 1984 ($n = 554$). There was considerable agreement between modeled and observed SSI values across the region and associated trends (e.g., Supplementary Fig. 7). The CanSWE dataset provides consistent April 1 SWE data from 1928–2020. For comparison against VIC modeled April 1 SWE, only CanSWE stations with at least 10 years of data after 1950 (the start of the VIC record) were selected ($n = 198$ in the study domain) and compared to the same model year April 1 SWE (Fig. 2c, d, diamonds above black Canadian border line). Differences in trends between VIC grid cells (6 km scale) and SNOTEL or CanSWE stations (point-scale) may exist due to the

differing spatial scales[23]. In this context, future work to explore snow water storage relationships and error using a finer spatial resolution model is needed.

**The Snow Storage Index (SSI).** The SSI represents differences in temporal offset and amplitude (i.e., magnitude) between precipitation seasonality and surface water input seasonality, creating a dimensionless value between −1 and 1. Long-term and annual SSI values were calculated by building on the methodology outlined in previous work[34], generating, first, a precipitation sine curve, subscript $P$, representing daily predicted precipitation:

$$P(t) = \bar{P}\left[1 + \delta_P \sin\left(\frac{2\pi(t - s_P)}{365}\right)\right] \quad (1)$$

where $\bar{P}$ is mean annual precipitation (mm), $t$ is the timestamp (days), $s_P$ is a precipitation phase shift (days), and $\delta_P$ is the dimensionless seasonal amplitude of precipitation. Second, a surface water inputs (SWI) sine curve was developed, subscript $SWI$, representing daily predicted surface water inputs, which are the rainfall and snowmelt, per day:

$$SWI(t) = \overline{SWI}\left[1 + \delta_{SWI} \sin\left(\frac{2\pi(t - s_{SWI})}{365}\right)\right] \quad (2)$$

where $\overline{SWI}$ is mean annual surface water inputs (mm), $t$ is the timestamp (days), $s_{SWI}$ is a surface water input phase shift (days), and $\delta_{SWI}$ is the dimensionless seasonal amplitude of surface water inputs. Surface water inputs are different from surface water or streamflow volumes, instead they represent only the inputs of water from the atmosphere as rainfall and snowmelt[25–27]. This definition is intended to capture the difference in timing between precipitation and associated water inputs to the terrestrial system. To generate the Snow Storage Index (SSI), the determined phase and amplitude from Eqs. 1 and 2 are mathematically related using a similarity index[34]:

$$SSI = -\left[\delta_{SWI} \, \text{sgn}(\delta_P) \cos\left(\frac{2\pi(s_{SWI} - s_P)}{365}\right)\right] \quad (3)$$

where the SSI describes whether or not surface water inputs are in phase with precipitation, and is weighted by the respective amplitudes. The final calculations were multiplied by −1 to emphasize typical snow-water-storing behavior as a positive value. The SSI evaluation in this work was constrained to areas, within the described ecoregions, with a long-term average SSI greater than 0 (SSI = 0 indicates uniformity in precipitation and surface water inputs, and SSI = 1 indicates a complete phase shift between precipitation and surface water inputs). This methodology thus assumes that grid cells showing a substantial delay between precipitation and surface water inputs are snow-dominated, and we focus on these areas. Examples of input data and resultant SSI values are shown in Fig. 1b, c.

The use of sine curves within the SSI calculation to fit and predict daily precipitation data has been done in the past, particularly in snow-dominated regions[34], with the intent of summarizing and grouping domains as: strongly winter-dominant precipitation, uniform precipitation, and strongly summer-dominant precipitation[33]. More often, the seasonality of precipitation in mountainous, western North America appears sinusoidal, with a peak in precipitation in winter months and an equivalent trough in summer months.

Hence, this methodology typically generates a high-quality fit when using long-term precipitation and surface water input data, where the two variables are first averaged over the record (1950–2013) prior to generating a single, long-term average SSI value (normalized residual sum-of-squares, RSS, shown in Supplementary Fig. 8). Larger normalized residuals, intended to portray the quality of the sine curve fit to long-term precipitation and surface water input data, appear to exist in regions where the associated sine curve has a larger amplitude. However, Fig. 1b, c do not suggest that this methodology is more or less appropriate for areas with more seasonal precipitation or surface water input generation, given the dynamic nature of the sine curve to a shift in phase and amplitude between highly seasonal to uniform data. Thus, a sine curve fit to less seasonal precipitation or surface water input data will similarly reflect the temporal patterns of these variables and appropriately result in a lower SSI.

These lower SSI values, closer to 0 (the threshold used in this work), can occur with more than one hydrologic condition: the amplitude of the sine curves (i.e., the seasonality of precipitation or surface water inputs) may be low, the phase (i.e., timing) of the precipitation curve is shifted toward spring months, or the phase of the surface water input curve is shifted toward winter months. Despite the different reasons for a lower (0–0.5) versus a greater (>0.75) SSI value, the metric is consistent in that it represents the degree to which snow is delaying the timing (and magnitude) of surface water input relative to precipitation. Thus, a lower SSI value (0–0.5), even when occurring in mountainous regions that store water as seasonal snow, reveals that the timing and magnitude of water inputs in this area is less strongly dictated by snow water storage than an area with a greater (>0.75) SSI value, carrying implications for changes in hydrology under current and future climate conditions. Representing the specific source of decreased SSI is beyond the scope of this work, aside from the examples used in Figs. 1c, 2b, 4 and S6.

Finally, in generating individual annual SSI values, daily precipitation and surface water input data from each water year were used and were noisier, resulting in a varying fit of each annual sine curve. In some instances, if a single month was

anomalously large in precipitation or surface water input amount, the peak of the sine curve underestimated the peak of the data. In cases where precipitation or surface water input showed bimodal patterns, the sine curve did not match at least one of the peak values. Thus, there were instances where a sine curve was not the best fit line for the annual data. Yet, annual data are required to evaluate the change in SSI through time, thus we validated the use of annual SSI calculations by assessing monthly precipitation, surface water inputs, snowmelt and rainfall trends (see section below). Changes in surface water inputs, snowmelt, and rainfall (Figs. 4 and S6) revealed a first principle explanation for the estimated decreases in snow water storage (i.e., the SSI), independent of the sine curve fit and SSI calculations, confirming that the sine curve methodology of the SSI does an adequate job of capturing snow water storage.

**Evaluating trends in the Snow Storage Index**. To evaluate the trends in the Snow Storage Index through time, Mann-Kendall tests were completed, and the 95% confidence threshold was applied[72,73]. These tests were completed at the grid cell scale, ecoregion scale, and as an average of western North America through the 64-year record (1950–2013). The slopes of these trends at the grid cell scale are shown in Fig. 2. When averaging SSI values, grid cells were weighted by area, which differed slightly depending on latitude[32]. Future work using the SSI will build intuition around SSI changes (unitless) and should consider related percent change.

**Evaluating drivers of the Snow Storage Index**. To evaluate the primary drivers of the SSI across all mountainous western North America ecoregions (Fig. 1a), an evaluation of precipitation, surface water inputs, and its components (i.e., rainfall and snowmelt) in areas where SSI was significantly changing were completed. An average total amount (depth in mm), for each of the four variables, was calculated per month. We then conducted a trend analysis of each variable through the 64-year record, regressing the variable of interest (dependent variable) by year (independent variable). This evaluation provided strong rationale for changes in the SSI, as the SSI is a function of precipitation and surface water input magnitude and seasonality. The analysis was first done using a 30-day moving window for each day of year so as not to be subject to the arbitrary constraints of the calendar month. The 30-day moving window analysis did not differ considerably from a calendar-month analysis and therefore it was determined that evaluating these variables by calendar month was suitable in providing a first principle explanation for a decline in SSI.

To evaluate the relationship between the annual SSI value (dependent variable) and a list of environmental variables across individual ecoregions, Pearson correlations were completed. The independent variables used in this analysis included seasonal and annual total precipitation, seasonal and annual average temperature, and annual average snow fraction. Each variable, including the SSI, was spatially averaged across each ecoregion and compared through time (across the 64-year record). Precipitation seasonality, melt fraction[10], peak and annual SWE, elevation, and potential evapotranspiration (PET) were also considered in this assessment but showed no significant relationships with the SSI, per ecoregion. The variables that showed statistically significant correlations ($p < 0.05$) are shown in Fig. 5 for annual, fall (September, October, November), winter (December, January, February), and spring (March, April, May) timeframes. Summer precipitation and temperature were evaluated against SSI and showed no statistically significant correlations and are thus not shown.

To further evaluate the sensitivities of the annual SSI to variability in temperature and precipitation, grid cell annual SSI anomalies were compared to corresponding precipitation and temperature anomalies at annual and seasonal timeframes. This methodology followed that outlined in previous, related work[10], where all annual anomalies (SSI, precipitation, and temperature) were calculated as the difference from the respective 64-year mean. Seasonal mean temperature and total precipitation were evaluated following the same monthly clusters as in Fig. 5. To characterize the relative influence of precipitation and temperature on the SSI across the anomaly space, the data were divided into SSI anomaly percentile bins. Given the very large number of data points ($n = 881,920$ grid cell years, i.e., 13,780 grid cells × 64 years), an average of each SSI anomaly percentile was derived. This reduced the number of points to six, corresponding to the 10th, 30th, 50th, 70th, 90th, and 99th percentiles. A centroid representing mean temperature and precipitation anomalies per SSI anomaly percentile bin was computed for each of the six bins. We then compared SSI anomaly percentile bins to seasonal fractional precipitation. This analysis was done as a combination of all grid cells across the domain and within their individual ecoregion (gray lines across all panels in Fig. 6). Because precipitation and temperature showed negligible effects on SSI in summer months, due primarily to relatively low precipitation totals during this time, we did not include the summer season in Fig. 6.

## Data availability
To access the formatted long-term average and annual (1950–2013) Snow Storage Index products; a formatted version of the meteorological forcing datasets and modeled flux datasets, a formatted version of the observational datasets, and the primary code used in this manuscript please visit: https://doi.org/10.5061/dryad.3bk3j9kpn. Meteorological data used to force the Variable Infiltration Capacity (VIC) model across North American

from 1950-2013 and the resulting flux data were downloaded from the NOAA National Centers for Environmental Information (https://www.ncei.noaa.gov/access/metadata/landing-page/bin/iso?id=gov.noaa.nodc:Livneh-Model). Observational SNOTEL data were downloaded from the Pacific Northwest National Laboratory (https://dhsvm.pnnl.gov/bcqc_snotel_data.stm), and the Canadian historical Snow Water Equivalent dataset was downloaded from the following Zenodo repository[31]. Level III eco-region boundaries for mapping and analysis were accessed from the EPA (https://www.epa.gov/eco-research/level-iii-and-iv-ecoregions-continental-united-states).

## Code availability
To access the R code used to generate the modeled and observed long-term average and annual Snow Storage Index products and associated metrics and figures in this manuscript please visit: https://doi.org/10.5061/dryad.3bk3j9kpn.

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

## Acknowledgements

This research has been supported by the National Aeronautics and Space Administration [NNX17AF50G, 80NSSC17K0071, 80NSSC20K1420], the National Oceanic and Atmospheric Administration [NA15OAR4310144], and the Western Water Assessment [NA21OAR4310309].

## Author contributions

K.E.H. developed the concept of the study together with N.P.M. who provided regular feedback and discussion for data analysis, results, and manuscript revisions. K.N.M. provided periodic feedback around project development, K.S.J. provided the code

incorporating precipitation phase partitioning in the observed dataset, and B.L. provided the primary modeled dataset required to complete the analysis. K.E.H. performed the analysis and prepared the first draft of the manuscript. All co-authors provided recommendations for the data analysis, participated in discussions about the results, and edited the manuscript through its final draft.

## Competing interests

The authors declare no competing interests.
