## [Peer Review File · Communications Earth & Environment]

7th Oct 22

Dear Dr Hale,

Please allow us to apologise for the delay in sending a decision on your manuscript titled "Recent decreases in snow water storage in western North America". It has now been seen by 3 reviewers, and I include their comments at the end of this message. They find your work of interest, but some important points are raised. We are interested in the possibility of publishing your study in Communications Earth & Environment, but would like to consider your responses to these concerns and assess a revised manuscript before we make a final decision on publication.

We therefore invite you to revise and resubmit your manuscript, along with a point-by-point response that takes into account the points raised. Please highlight all changes in the manuscript text file. In particular, please consider carefully the following elements:

- better highlight the advantages/disadvantages and limitations of using the new SSI metric compared to more traditional approaches (eg. other common snow metrics like SWE or snow cover duration, or other drought indices developed elsewhere as commented by reviewers)
- Better explain the derivation of the SSI index and its applicability to non-seasonal precipitation regimes
- consideration of ground/soil water storage effects, as well as snow sublimation, in the analysis and metric definition, or include these aspects in the discussion

Please use the following link to submit your revised manuscript, point-by-point response to the referees' comments (which should be in a separate document to any cover letter) and the completed checklist:

[link redacted]

We hope to receive your revised paper within six weeks; please let us know if you aren't able to submit it within this time so that we can discuss how best to proceed. If we don't hear from you, and the revision process takes significantly longer, we may close your file. In this event, we will still be happy to reconsider your paper at a later date, as long as nothing similar has been accepted for publication at Communications Earth & Environment or published elsewhere in the meantime.

We understand that due to the current global situation, the time required for revision may be longer than usual. We would appreciate it if you could keep us informed about an estimated timescale for resubmission, to facilitate our planning. Of course, if you are unable to

estimate, we are happy to accommodate necessary extensions nevertheless.

Please do not hesitate to contact me if you have any questions or would like to discuss these revisions further. We look forward to seeing the revised manuscript and thank you for the opportunity to review your work.

Best regards,

Christophe Kinnard, PhD
Editorial Board Member
Communications Earth & Environment
orcid.org/0000-0002-4553-5258

Joe Aslin
Senior Editor
Communications Earth & Environment

EDITORIAL POLICIES AND FORMATTING

Editorial Policy: [Policy requirements](https://www.nature.com/documents/nr-editorial-policy-checklist.pdf) (Download the link to your computer as a PDF.)

Furthermore, please align your manuscript with our format requirements, which are summarized on the following checklist:

[Communications Earth & Environment formatting checklist](https://www.nature.com/documents/commsj-phys-style-formatting-checklist-article.pdf)

and also in our style and formatting guide [Communications Earth & Environment formatting guide](https://www.nature.com/documents/commsj-phys-style-formatting-guide-accept.pdf) .

***** DATA:** Communications Earth & Environment endorses the principles of the Enabling FAIR data project (<http://www.copdess.org/enabling-fair-data-project/>). We ask authors to make the data that support their conclusions available in permanent, publically accessible data repositories. (Please contact the editor if you are unable to make your data available).

All Communications Earth & Environment manuscripts must include a section titled "Data Availability" at the end of the Methods section or main text (if no Methods). More information on this policy, is available at <http://www.nature.com/authors/policies/data/data-availability-statements-data-citations.pdf>.

If a community resource is unavailable, data can be submitted to generalist repositories such as [figshare](https://figshare.com/) or [Dryad Digital Repository](http://datadryad.org/). Please provide a unique identifier for the data (for example a DOI or a permanent URL) in the data availability statement, if possible. If the repository does not provide identifiers, we encourage authors to supply the search terms that will return the data. For data that have been obtained from publically available sources, please provide a URL and the specific data product name in the data availability statement. Data with a DOI should be further cited in the methods reference section.

REVIEWER COMMENTS:

Reviewer #1 (Remarks to the Author):

Review of “Recent decreases in snow water storage in western North America” by Hale et al.

by Michal Jenicek

General comments

Authors characterized volume and temporal offset between precipitation and surface water inputs (SWI) across twelve ecoregions in western United States and southwest Canada over the period 1950-2013. For this, they defined a new metric, Snow storage index (SSI) which represents differences in temporal offset and magnitude between precipitation seasonality and surface water input seasonality. The research question behind was “How does the timing and magnitude of snow water storage vary across western North America and how has it changed in recent decades?” Authors found that SSI declined mostly in Columbia Mountains,

Canadian Rockies, (Northern) Cascades and partly Southern Rockies as a combination of different reasons, e.g., earlier snowmelt and rainfall in spring months or winter precipitation decrease. Authors found that the SSI is a useful metric of snow water storage that is applicable to both hydrological and ecological processes and thus can be used for the quantification of the temporal offset between precipitation and surface water inputs.

In my opinion, authors did an interesting work which has high scientific relevance. I agree with authors that the use of SSI as a metric describing both magnitude and temporal offset of precipitation and SWI is novel. Although the results mostly confirm our existing knowledge regarding changes in snow storages and water inputs, I found the study important and novel, thus appropriate for Communications Earth & Environment. I have only a few (rather minor) comments listed below, which I would like to be addressed before I can recommend the manuscript for publication.

Specific comments

While I agree with most of analyses and interpretation, I am not fully convinced that using the new metric is better than analysing mutual relations between snow and climate characteristics, such as SWE, snow cover duration and snowmelt timing, all of them reflecting changes in air temperature and precipitation. Does the new metric improve our understanding about the processes? While I see the advantage of the new metric in combining of both magnitude and timing of SWI into one single metric, for the reader is not always easy to understand all shortcomings and potential consequences stemming from its calculation (see also my next comment). Therefore, it might be sometimes easier to use “traditional” metrics because they are easy to understand, and they have a clear physical background. Although I appreciate that authors gave a lot of effort in explaining the new metric in detail, I would encourage them to better highlight advantages/disadvantages and limitations in using the new metric compared to more traditional approaches (see also my bellow comments).

L 202, Fig. 2 a, b: The largest decrease in SSI was found in Columbia Mountains/Northern Rockies and Canadian Rockies (although mostly not confirmed by observed data). Authors mentioned that the reason was decreased winter precipitation which resulted in “decreased amplitudes of the related sine curves, which reduced SSI” (L 202). Thus, the reason was not the decrease in winter precipitation itself, but rather the resulting decrease in precipitation seasonality. If this my interpretation is correct, isn't it a clear limitation of using SSI only for highly seasonal precipitation climates? Although I found the SSI as useful metric, its decrease/increase does not always mean a change in hydrological behaviour of catchments. For example, the decreasing precipitation seasonality does not necessarily mean that the seasonal distribution of water inputs is changing as well. Besides some more discussion on this topic, authors can also consider adding a point 3) in lines 132-133 where “decreasing precipitation seasonality” (or similar) will be mentioned next to existing two points.

From the above comment, perhaps more general question arises, namely to what extent is the SSI usable in other world regions, especially in those with relatively unique distribution of precipitation during the year and thus low precipitation seasonality (e.g., most of European Alps)? Please discuss shortly.

Authors identified earlier snowmelt and thus SWI as one of the reasons of declining SSI. May also later snow cover onset influence the resulting SSI? While I would say that not much (to my knowledge most of the studies did not find trends in snow cover onset in last decades), I would appreciate some discussion on this topic.

With increasing air temperature and thus increasing amount of rain during winter compared to snow (decreasing snowfall fraction), one would expect increase in partial snowmelt (and thus SWI) during winter and spring months due to increasing rain-on-snow events (at least for some areas and elevations). How could the SSI be influenced by the increase in winter?

Technical corrections

Abstract: In my opinion, “quantifying snowpack water storage in terms of its’ magnitude and duration” is not ignored as formulated by authors since I think several studies exist which deal with this issue (although maybe not many). I therefore suggest reformulating the sentence to soften this formulation a bit.

Abstract and several places in the text: Authors are sometimes mixing terms “snowfall/snow fractions”, although I think that “snowfall fraction” is always meant. Please unify and use consistently throughout manuscript.

Please consider increasing axis captions in Figs. 3, 4, S1, S2 and S7.

Reviewer #2 (Remarks to the Author):

The manuscript “Recent decreases in snow water storage in western North America” by Hale and co-authors presents a new definition of a snow storage index (SSI), as well as a thorough analysis of VIC simulations with the new metric. The basic premise of the paper is important and does provide some novelty to the literature: how can snow storage – both temporally and in terms of magnitude – be jointly quantified into a single metric so that it can be tracked in space in time. Personally, I find the temporal evolution of a metric like SSI to be very powerful, and hopefully a useful indicator to show policy makers or other decision makers how anthropogenic warming is impacting water resources. Furthermore, the paper is well written and clear, with only minor suggestions from me on how to improve the presentation of the paper (see below). Overall, job well done.

However, my main issue is with how “surface water inputs” are defined and treated in this context (see below) and the apparent disregard of a major component of the hydrologic water cycle: groundwater and soil water storage. This is more than just semantics in their nomenclature of SWI (although renaming it to more aptly snow surface water inputs would help), as many of their main results are derived from a set of simulations which are neglecting the pivotal role which groundwater flow has on streamflow. Surely this cannot be overlooked in what could be a very nice contribution to the literature. As such, I encourage the authors to think deeply about how to incorporate this weak point of their analysis and

include further discussion of the implications in the manuscript (not just in the response to reviewer comments).

Major

1. Line 36 – From the beginning here, I have issue with how “surface water inputs” is being introduced here, “representing the sum of rainfall and snowmelt.” Are the authors completely ignoring groundwater inputs to surface water in the definition?

Line 291: How can surface water inputs only be defined as snowmelt plus rainfall?

Fundamentally there is a complete disregard of groundwater to streamflow, which, especially for many Mediterranean climate regions of the western US, are driven by groundwater inputs during baseflow.

At a minimum, I would suggest the authors rename the surface water input (line 340 variable) and associated discussions to instead be the snow surface water input, which is actually what they are defining. But moreover I would encourage the authors to include some discussion of how this definition is potentially limiting, acknowledge the depth of literature on the connection between snowmelt, groundwater, and streamflow, and/or think about how groundwater inputs to surface water can be incorporated into their SSI definition.

2. Line 305 – The authors allude to the frozen soil option not being selected so that there is no subsurface storage and to ensure a conservative estimate of spring runoff, but this is not what’s occurring in the physical world. Additionally, much of the spring snowmelt will end up in soil and deeper groundwater storage (depending on conditions), which as I describe above, does not seem to be accounted for in their approach.

3. Line 366 – The fact that the SSI metric can give the same answer for different reasons (different degrees of precipitation or snow surface water inputs, earlier or later precipitation/snow surface water inputs) makes me think that there could (or should) be different sub-definitions of the SSI. That way, two classifications with the same value of the metric, can be differentiated for the root causality, which is potentially very important given the physical drivers of the hydrology. I think this is a missed opportunity of the authors to expand on more.

The authors get at this point in their discussion a little (line 132 and 190) but only through examples of two locations. Maybe it’s outside of the scope of this paper to include new subsets of definitions here, but since they are putting forth a new definition, it seems appropriate to at least discuss what these different “flavors” of SSI could/should include.

4. How different are the sine curve definitions of the peak SWI and P compared to the actuals (i.e. the day at which those variables peaked)? In the example show in Figure 1c, for example, the SWI has an outlier peak in April, where it’s visually clear that the seasonal peak occurs closer to June – that’s a good thing. But how representative are these examples of other years, other grid-cell behavior?

5. Line 79 – phrasing the motivation of snowpack and water storage around warming changes alone is too simplistic. What about all the other perturbations and expected changes in extremes and secondary effects (e.g. vegetation shifts, wildfires, etc.)? Consider re-wording here and throughout.

6. Line 116 – Most of the Colorado front range on your map has negative SSI. Similarly, only a very small fraction of the Sierra Nevada on Fig 1a has high SSI. While I like the comparison and the reference here, I think a direct comparison of those regions in their entirety is a stretch.

7. Fig 2c-d – Can you provide a possible explanation for the positive changes in SSI at some of these gauges? Furthermore, while line 147 says 80% of SNOTEL show a decrease, it's very hard to see on these graphs. Is the change minimal (yellow colors)? Also, much work on SNOTEL changes has been done by others (see work by Mote and co-authors for example), please consider referencing.

It's also hard to infer what a 0.1 change versus a 0.2 change in SSI really means. Perhaps if this metric becomes more common, inference about if these differences are/are not substantial would come along with it, but as a reader just introduced to the metric, I find it hard to put into perspective.

8. Figure S4 – If the Canadian Rockies were excluded, is there still a decline in the western North American SSI decline? The statistically significant cells are few and far between within the CONUS if the northern most regions were neglected, making this almost most applicable to a regional phenomena.

9. Figure 6 – It's not clear what this graph is plotting. What are the gray lines? Are these a select few anomalously wet or dry years, with the average superimposed with a SSI dot? It's not evident from the paragraph of text or the caption and needs further clarification.

Minor

1. Line 39 - the two sentences starting with "Different from..." could appear earlier in the abstract, perhaps after the SSI is first mentioned to provide a better introduction to the topic.

2. Line 67 - can you define or provide some examples of what you mean by snow water storage dynamics? Do you mean here the amount of water stored in snow? The word "dynamics" makes it sound as though you're referring to some processes.

3. Fig 2b these insets are really hard to see as they are so small.

4. Figure 3 – figure quality is very poor, consider rotating to landscape to increase dpi.

Reviewer #3 (Remarks to the Author):

This paper introduces an interesting approach to quantify the hydrologic significance of the snowpack, the snow storage index (SSI). The SSI is used to conduct a trend analysis in North west America, revealing regions where climate change had an significant impact on the snowmelt contribution to runoff. The paper reads well. The conclusions are supported by a large dataset of ground measurements and the output from a model simulation that was evaluated in a previous study. I have only a few concerns and some suggestions.

- In the supplement it is written that "Snowmelt was calculated as the negative change in daily SWE". Why was sublimation neglected?

- "We averaged modeled and observed SSI as both an arithmetic average (average of grid-cell SSI values) and an average of its constituents (average of area-wide precipitation and surface

water inputs)". Later it is written that "We employed a gridded hydrometeorological dataset that contains spatially and temporally continuous daily meteorological forcings and simulated Variable Infiltration Capacity (VIC) model states and fluxes at 1/16°". The spatial averaging should account for the fact that the VIC grid is not equal area (otherwise high latitude cells will have too much weight). I could not check if this was actually done since the code to produce the analyses was not made available by the authors.

- Surprisingly, the derivation of the SSI formulae is not well explained. The authors fit both P and SWI time series using a sine function but then it is only mentioned that "the two sine curves were combined" but what does "combine" exactly mean?

- Since the paper aims to introduce a new index, I think it would be useful to discuss how it complements previous indices, which also aimed to represent the hydrologic role of snow such as:

Staudinger et al. (2014), A drought index accounting for snow. WRR

Huning & Aghakouchak (2020), Global snow drought hot spots and characteristics. PNAS

Mankin et al. (2015) The potential for snow to supply human water demand in the present and future. ERL

In particular, the "snow resource potential" in the Mankin study seems quite similar to the SSI (if yes revise manuscript line 252).

- L147, "Observed SSI trends across the area (Fig. 2c and d) corroborate the model-based analysis" But looking at fig. 2, model-based and station-based SSI changes do not look really consistent?

- L292, the authors emphasized the uncertainty associated to the rain-snow partition temperature. But why focusing on this aspect? Many other parameterizations could cause model errors. The 6 km resolution of the model is probably a major source of error in high mountain regions, because the model forcing are assumed to be homogeneous over an area with a large meteorological variability (including the air temperature which is used to estimate the rain-snow fraction).

Two (optional) suggestions to better show the value of the SSI for future studies:

- High SSI values should be found in mountainous regions under the influence of the Mediterranean climate (cold and wet winter, dry and hot summer, Fayad et al. 2017 JoH). It would be interesting to investigate how the SSI patterns relate to a more standard climate classification (e.g. Koppen).

- It would be interesting to evaluate if the SSI trends spatially match river flow centroid trends across North America (e.g. Stewart et al. 2005 or a more recent study if available).

Response to reviewers:

We thank the reviewers for their time and insightful comments and suggestions. We have addressed each point below using green, indented text. References to line numbers correspond to the new manuscript draft.

In summary, we now provide additional explanations to using Snow Storage Index across any region (e.g., regions within in this study or others in the future), including areas with non-seasonal precipitation regimes. We also better highlight the metric's advantages, disadvantages, and limitations. Our revisions include:

- A more explicit distinction between surface water inputs and surface water.
- A more direct comparison of the SSI to other metrics traditionally used to characterize snow water resources and predict streamflow.
- New introductory and discussion sentences on ground/soil water storage effects on water availability and its tangential relationship with the SSI.
- Improved description of analysis methods regarding spatial averaging and sublimation.
- Updated figures.

Below are our general and line-by-line responses to all three reviewers' comments.

Reviewer #1:

Authors characterized volume and temporal offset between precipitation and surface water inputs (SWI) across twelve ecoregions in western United States and southwest Canada over the period 1950-2013. For this, they defined a new metric, Snow storage index (SSI) which represents differences in temporal offset and magnitude between precipitation seasonality and surface water input seasonality. The research question behind was "How does the timing and magnitude of snow water storage vary across western North America and how has it changed in recent decades?" Authors found that SSI declined mostly in Columbia Mountains, Canadian Rockies, (Northern) Cascades and partly Southern Rockies as a combination of different reasons, e.g., earlier snowmelt and rainfall in spring months or winter precipitation decrease. Authors found that the SSI is a useful metric of snow water storage that is applicable to both hydrological and ecological processes and thus can be used for the quantification of the temporal offset between precipitation and surface water inputs.

In my opinion, authors did an interesting work which has high scientific relevance. I agree with authors that the use of SSI as a metric describing both magnitude and temporal offset of precipitation and SWI is novel. Although the results mostly confirm our existing knowledge regarding changes in snow storages and water inputs, I found the study important and novel, thus appropriate for Communications Earth & Environment. I have only a few (rather minor) comments listed below, which I would like to be addressed before I can recommend the manuscript for publication.

We thank the reviewer for this overall assessment, and we are glad that the reviewer sees the contribution of this targeted work to broader water resources readership and scientific communities.

While I agree with most of analyses and interpretation, I am not fully convinced that using the new metric is better than analyzing mutual relations between snow and climate characteristics, such as

SWE, snow cover duration and snowmelt timing, all of them reflecting changes in air temperature and precipitation. Does the new metric improve our understanding about the processes? While I see the advantage of the new metric in combining of both magnitude and timing of SWI into one single metric, for the reader is not always easy to understand all shortcomings and potential consequences stemming from its calculation (see also my next comment). Therefore, it might be sometimes easier to use “traditional” metrics because they are easy to understand, and they have a clear physical background. Although I appreciate that authors gave a lot of effort in explaining the new metric in detail, I would encourage them to better highlight advantages/disadvantages and limitations in using the new metric compared to more traditional approaches (see also my bellow comments).

We agree with the reviewer that it is important to clearly state the advantages and disadvantages of this new metric, particularly for an audience that may be more accustomed to independently assessing traditional snow metrics against climate characteristics. The importance of water storage in the snowpack is the role it plays in altering the timing and rate of water delivery into terrestrial systems. In essence, this is the only differentiation of snowfall from rainfall in terms of hydrology. The SSI shows how the time-magnitude duration of snow water storage can change, unlike a trend in SWE which may only capture a snapshot in time (e.g., April 1). In response to this comment, we have added the following text in the manuscript:

“The use of a single metric to assess trends in both the temporal differences and relative magnitudes of precipitation and corresponding surface water inputs, as rainfall and snowmelt, across regions has critical implications to better monitor ecosystem stress and inform water resource management [Kiewiet et al., 2022; Kormos et al., 2014; Niemeyer et al., 2016].” (Lines 72-76)

“Importantly, the SSI metric quantifies the temporal offset between precipitation and surface water inputs, in addition to the amount of water stored in the snowpack, a combination of hydrologic characteristics not previously captured in a single value. The SSI complements preceding indices aimed at quantifying the integrative hydrologic impact of seasonal snow, such as the temporal lag between precipitation and snowmelt used to identify downstream hydrologic deficits from snowmelt and rainfall [Staudinger et al., 2014] and the potential for snowmelt to meet downstream demands in hydrologically vulnerable regions of the world [Immerzeel et al., 2020; Mankin et al., 2015].” (Lines 276-283)

L 202, Fig. 2 a, b: The largest decrease in SSI was found in Columbia Mountains/Northern Rockies and Canadian Rockies (although mostly not confirmed by observed data). Authors mentioned that the reason was decreased winter precipitation which resulted in “decreased amplitudes of the related sine curves, which reduced SSI” (L 202). Thus, the reason was not the decrease in winter precipitation itself, but rather the resulting decrease in precipitation seasonality. If this my interpretation is correct, isn't it a clear limitation of using SSI only for highly seasonal precipitation climates? Although I found the SSI as useful metric, its decrease/increase does not always mean a change in hydrological behavior of catchments. For example, the decreasing precipitation seasonality does not necessarily mean that the seasonal distribution of water inputs is changing as well. Besides some more discussion on this topic, authors can also consider adding a point 3) in

lines 132-133 where “decreasing precipitation seasonality” (or similar) will be mentioned next to existing two points.

The reviewer is correct that the decrease in seasonality, by way of decreased winter precipitation led to a decrease in the SSI (see Figure S6 of the current manuscript). We have corrected the following statements for clarity:

“2) declines in precipitation seasonality.”(Lines 139)

“In some northern ecoregions, predominant decreases in precipitation seasonality, by way of decreased winter precipitation, resulted in decreased amplitudes of the related precipitation sine curves, which reduced SSI (Fig. 2b.ii).” (Lines 209-211)

“In select ecoregions (e.g., Canadian Rockies and Columbia Mountains/Northern Rockies), there exists a secondary mechanism for SSI declines, which is a declining trend in winter precipitation and thus annual precipitation seasonality (Fig. S6) [DeBeer et al., 2016].” (Lines 259-262)

However, respectfully, the reviewer is mistaken in their interpretation that this is a shortcoming of the SSI. If precipitation seasonality is low, the SSI will remain closer to 0 than in areas with higher precipitation seasonality. Yet, year-to-year changes in precipitation seasonality are captured by the dynamic precipitation sine curve. In all ecoregions, the timing or magnitude of surface water inputs changed in some way. In the Canadian Rockies and Columbia Mountains/Northern Rockies, the decrease in winter precipitation and thus precipitation seasonality was the **predominant** driver of decreased SSI. However, in these ecoregions there was **also** a shift toward earlier surface water inputs (revised Fig. S6 pasted below). Thus, this work quantifies all changes of the ability of western North American mountains to store water as snow and delay water inputs to the terrestrial system, regardless of the process behind the signal. In response to this comment, we have revised this statement:

“Specifically, precipitation in December, January, and February has significantly decreased in the Canadian Rockies and Columbia Mountains/Northern Rockies which, combined with earlier snowmelt, significantly decreased surface water inputs in June and July (Fig. S6).” (Lines 211-214)

Precip (mm)

SWI (mm)

Melt (mm)

Rain (mm)

Fig. S6: Monthly precipitation, surface water inputs, snowmelt, and rainfall in Canadian Rockies and Columbia Mountains/Northern Rockies ecoregions. Trend lines indicate $p < 0.05$.

From the above comment, perhaps more general question arises, namely to what extent is the SSI usable in other world regions, especially in those with relatively unique distribution of precipitation during the year and thus low precipitation seasonality (e.g., most of European Alps)? Please discuss shortly.

The SSI will characterize the precipitation and surface water input (SWI) temporal distribution of any region, representing the extent to which a region stores and later releases water into a terrestrial system. While the European Alps have considerable snowfall and thus considerable snowmelt derived-SWI, the more uniform precipitation seasonality will decrease the local to regional SSI. Further, population centers of Europe do not rely on snowpack water relative to places that receive zero summer precipitation. In response to this comment, we have added the following:

“The hydrology of areas experiencing lower SSI values (0-0.5) by way of less seasonal precipitation and/or reduced surface water inputs may exhibit less hydrologic sensitivity to changes in snowpack water storage associated with warming.” (Lines 270-273)

Authors identified earlier snowmelt and thus SWI as one of the reasons of declining SSI. May also later snow cover onset influence the resulting SSI? While I would say that not much (to my knowledge most of the studies did not find trends in snow cover onset in last decades), I would appreciate some discussion on this topic.

The reviewer is correct that decreases in autumn snowfall or later snow cover onset (potentially due to decreased snowfall and/or increased snowmelt derived-SWI shortly after snowfall) would result in a decrease in the SSI. Any climatic change that more closely aligns the timing of precipitation inputs with surface water input generation will result in a decrease in SSI. Examples of increases in fall season SWI were limited. However, the nuance is important and further exemplifies the movement of the SSI metric. In response to this comment, we have added the following sentence:

“Fall season precipitation shifts from snowfall to rainfall and/or increases in surface water inputs by way of early season melt would similarly decrease SSI. This fall season signal was not explicitly evaluated herein as the climate sensitivities during this period were modest relative to spring and early summer.” (Lines 288-291)

With increasing air temperature and thus increasing amount of rain during winter compared to snow (decreasing snowfall fraction), one would expect increase in partial snowmelt (and thus SWI) during winter and spring months due to increasing rain-on-snow events (at least for some areas and elevations). How could the SSI be influenced by the increase in winter?

The reviewer is correct that increases in SWI by way of increased rain-on-snow events would decrease the SSI. SWI would more closely follow the seasonality of precipitation under these conditions. This would decrease the seasonality of SWI and thus the amplitude of the sine curve fit to these data. In response to this comment, we have added the following:

“Winter or early spring rain-on-snow events would also act to align precipitation and surface water input timing and thus decrease the SSI.” (Lines 291-293)

Technical corrections:

Abstract: In my opinion, “quantifying snowpack water storage in terms of its’ magnitude and duration” is not ignored as formulated by authors since I think several studies exist which deal

with this issue (although maybe not many). I therefore suggest reformulating the sentence to soften this formulation a bit.

We thank the reviewer for this suggestion. We have removed this sentence and believe the change softens the language but maintains the point of novelty.

Abstract and several places in the text: Authors are sometimes mixing terms “snowfall/snow fractions”, although I think that “snowfall fraction” is always meant. Please unify and use consistently throughout manuscript.

We thank the reviewer for this comment. We have combed through the manuscript and stated “*snowfall fraction*” when/where appropriate. Any other mention of snowfall is referring to falling snow from the atmosphere.

Please consider increasing axis captions in Figs. 3, 4, S1, S2 and S7.

We thank the reviewer for this suggestion. We have increased font size and/or reconfigured each of these figures to increase the readability of all figure components. See the reformatting of Fig S7 (now S6 in the revised manuscript) above, in addition to the following revised figures:

Fig. 3: Spatial average modeled SSI has significantly declined from 1950-2013. (a) Spatially weighted average VIC-based annual SSI with daily precipitation and surface water inputs averaged spatially prior to the SSI calculation (purple line), and with SSI spatially averaged by grid cell after each grid cell SSI was calculated (black line); standard errors are shown with purple and black shading, respectively. Trend lines for SSI through time are shown in both cases ($p < 0.01$), (b) Average modeled decadal SSI across the study region, showing the median, interquartile range and outliers across the 6 decades over the VIC record. Trend line indicates $p < 0.05$, with standard errors shown with black shading.

Fig. 4: Surface water inputs increase in spring and decrease in summer months. Averaged across grid cells where SSI is significantly declining, spatial average monthly precipitation, surface water inputs (SWI), snowmelt, and rainfall in (a) March and (b) July from 1950-2013; trend lines indicate $p < 0.05$.

Fig. S1: Histograms of long-term \overline{SSI} distribution. Histograms of long-term \overline{SSI} distribution in each ecoregion within the study domain (where $\overline{SSI} \geq 0$).

Fig. S2: Weak relationships between SSI versus SWE and elevation. Total annual SWE (mm) and elevation (m) were compared to long-term SSI and average-annual SSI in each major ecoregion within the study domain. Relationships are either insignificant ($p > 0.05$) or generate $r^2 < 0.3$.

Reviewer #2:

The manuscript “Recent decreases in snow water storage in western North America” by Hale and co-authors presents a new definition of a snow storage index (SSI), as well as a thorough analysis of VIC simulations with the new metric. The basic premise of the paper is important and does provide some novelty to the literature: how can snow storage – both temporally and in terms of magnitude – be jointly quantified into a single metric so that it can be tracked in space in time. Personally, I find the temporal evolution of a metric like SSI to be very powerful, and hopefully a useful indicator to show policy makers or other decision makers how anthropogenic warming is impacting water resources. Furthermore, the paper is well written and clear, with only minor suggestions from me on how to improve the presentation of the paper (see below). Overall, job well done.

However, my main issue is with how “surface water inputs” are defined and treated in this context (see below) and the apparent disregard of a major component of the hydrologic water cycle: groundwater and soil water storage. This is more than just semantics in their nomenclature of SWI (although renaming it to more aptly snow surface water inputs would help), as many of their main results are derived from a set of simulations which are neglecting the pivotal role which groundwater flow has on streamflow. Surely this cannot be overlooked in what could be a very nice contribution to the literature. As such, I encourage the authors to think deeply about how to incorporate this weak point of their analysis and include further discussion of the implications in the manuscript (not just in the response to reviewer comments).

We thank the reviewer this overall assessment and summary of the work. We are excited to hear that the reviewer agrees on the potential and quantitative power of the SSI for future, applied work. We appreciate the feedback regarding the definition of “surface water inputs” as well as discussing the role of ground/soil water contributions to surface water.

We agree with the reviewer that soil and ground water are essential components of the terrestrial hydrology system. However, the aim of this manuscript is to evaluate how the inputs of water **into** the soil are changing. As we define them, surface water inputs do not include soil or ground water storage or streamflow, but rather are the inputs of water to the terrestrial system from the atmosphere via rainfall or snowmelt. This is consistent with previous works: *Kiewiet et al., 2022, Kormos et al., 2014, Niemeyer et al., 2016*. Thus, we are not quantifying surface water in the stream network or the role of ground/soil water, which we expect play a minimal role in SWI generation and thus the SSI. To address the important point from the reviewer of soil and ground water storage and the distinction to surface water inputs, see the additions to the manuscript described and pasted within the following responses.

- Kiewiet, L., Trujillo, E., Hedrick, A., Havens, S., Hale, K., Seyfried, M., ... & Godsey, S. E. (2022). Effects of spatial and temporal variability in surface water inputs on streamflow generation and cessation in the rain–snow transition zone. *Hydrology and Earth System Sciences*, 26(10), 2779-2796.
- Kormos, P. R., Marks, D., McNamara, J. P., Marshall, H. P., Winstral, A., & Flores, A. N. (2014). Snow distribution, melt and surface water inputs to the soil in the mountain rain–snow transition zone. *Journal of Hydrology*, 519, 190-204.
- Niemeyer, R. J., Link, T. E., Seyfried, M. S., & Flerchinger, G. N. (2016). Surface water input from snowmelt and rain throughfall in western juniper: potential impacts of climate change and shifts in semi-arid vegetation. *Hydrological Processes*, 30(17), 3046-3060.

Major

Line 36 – From the beginning here, I have issue with how “surface water inputs” is being introduced here, “representing the sum of rainfall and snowmelt.” Are the authors completely ignoring groundwater inputs to surface water in the definition?

We thank the reviewer for this comment. In response, we have stated the definition of surface water inputs early in the manuscript to clarify that surface water inputs are not synonymous to surface water or streamflow (see specific manuscript changes below). Surface water inputs are only the inputs of water from the atmosphere via rainfall or snowmelt. In this way, groundwater or soil water storage is not considered. The explicit intent of this work is to focus on the role of snow as a lever to alter the timing and rate of water inputs from the atmosphere. The groundwater expression that the reviewer refers to is in relation to an input of surface water that is transferred from elsewhere in a watershed.

In response to this feedback, we have made changes to the text to clarify that we are not quantifying surface water or streamflow (changes in **bold**):

*“We introduce a Snow Storage Index (SSI), which represents the annual temporal phase difference between daily precipitation and surface water inputs – **sum of rainfall and snowmelt into the terrestrial system** – weighted by relative magnitudes.”* (Lines 31-34)

*“However, these variables do not describe the essential role that snowpack water storage plays in creating a temporal lag between precipitation inputs **and its availability to watersheds to eventually become streamflow, evapotranspiration, or soil/ground water.**”* (Lines 64-66)

“Surface water inputs, in the context of this work, represent the per grid-cell input of water to the terrestrial system from the atmosphere via rainfall or snowmelt only, and thus do not include lateral movement of soil water, ground water, or streamflow from one grid cell to another.” (Lines 79-82)

“Surface water inputs were assumed to be the summation of snowmelt and rainfall at each time stamp, distinctly different from quantifying surface water or streamflow, as we focused on water availability from the atmosphere at the terrestrial surface and not delivery downstream. Similarly, an evaluation of soil and ground water storage are also outside the scope of this work.” (Lines 347-351)

“Surface water inputs are different from surface water or streamflow volumes, instead they represent only the inputs of water from the atmosphere as rainfall and snowmelt [Kiewiet et al., 2022; Kormos et al., 2014; Niemeyer et al., 2016]. This definition is intended to capture the difference in timing between precipitation and associated water inputs to the terrestrial system.” (Lines 399-402)

Line 291: How can surface water inputs only be defined as snowmelt plus rainfall? Fundamentally there is a complete disregard of groundwater to streamflow, which, especially for many Mediterranean climate regions of the western US, are driven by groundwater inputs during baseflow.

Please see our response and changes associated with the previous comment.

At a minimum, I would suggest the authors rename the surface water input (line 340 variable) and associated discussions to instead be the snow surface water input, which is actually what they are defining. But, moreover, I would encourage the authors to include some discussion of how this definition is potentially limiting, acknowledge the depth of literature on the connection between snowmelt, groundwater, and streamflow, and/or think about how groundwater inputs to surface water can be incorporated into their SSI definition.

We thank the reviewer for this suggestion. We believe the clarification of the “surface water inputs” definition, based on the feedback above, provides a distinction from “surface water.” We believe “snow surface water input” is not an accurate term, as SWI also includes rainfall on snow or snow-free ground. This is a critical component to the SSI. Rainfall can greatly shift the seasonality of the SSI and influence the magnitude to which an area relies on the timing of SWI from snowmelt and influence the magnitude of sensitivity to changes in climate across western North America. The contribution of ground water and soil water into surface water is outside of the scope of this work and has been discussed in the lines below through the lens of future work:

“Future work should further examine how the decaying of mountain snow water storage manifests in the context of Earth’s ecosystems, water cycle, and water security.” (Lines 301-303)

“Similarly, while the effects of the subsurface are expected to be minimal with regards to surface water inputs, relevant connectivity among soil, groundwater, and vegetation should also be explored, particularly as it pertains to eventual streamflow [Cayan et al., 2010; McNamara et al., 2005].” (Lines 311-314)

Line 305 – The authors allude to the frozen soil option not being selected so that there is no subsurface storage and to ensure a conservative estimate of spring runoff, but this is not what’s occurring in the physical world. Additionally, much of the spring snowmelt will end up in soil and deeper groundwater storage (depending on conditions), which as I describe above, does not seem to be accounted for in their approach.

We thank the reviewer for their perspective on this issue. We wish to clarify that the model does indeed provide subsurface storage in all cases. The frozen soils option was not used in the chosen hydrologic model dataset because the authors of that dataset note that it represents a complex subroutine that introduces a number of new uncertainties related to calibration and the stability of the solver, etc. [Livneh et al., 2013; 2015]. However, and perhaps most importantly, the effect of frozen soils on SSI is expected to be negligible, since this option would primarily affect the fate of meltwater, e.g. whether water infiltrates into soils or directly becomes runoff, whereas the SSI is concerned with the snowpack itself and atmospheric water inputs (i.e., rainfall and snowmelt). We have revised the text to clarify the justification and expected impacts on SSI accordingly:

Original text: “A full iterative energy balance option was used within the Livneh et al. [2013; 2015] VIC simulation, while an explicit frozen soil option was not selected, to ensure a conservative estimate of spring runoff magnitude and rate, acknowledging that overestimating frozen soil effects could overstate linkages between snowmelt and runoff. This option represents realistic near-surface soil freezing dynamics across mountainous,

western North America [Zhang et al., 2003a-b], and the impact of frozen soils on SWE is likely minimal, while the evaluation of runoff/streamflow was not a part of this work.”

Revised text: “A full iterative energy balance option was used within the Livneh et al. [2013; 2015] VIC simulation, while an explicit frozen soil option was not selected given the additional uncertainties introduced by the complexity of the frozen soil routine and its numerical instabilities. The lack of frozen soils are expected to have negligible impact on SSI, since the SSI is primarily an ‘above ground’ metric, while the choice of frozen soil option would primarily affect the fate of meltwater, i.e., infiltration versus overland flow.” (Lines 360-365)

Line 366 – The fact that the SSI metric can give the same answer for different reasons (different degrees of precipitation or snow surface water inputs, earlier or later precipitation/snow surface water inputs) makes me think that there could (or should) be different sub-definitions of the SSI. That way, two classifications with the same value of the metric, can be differentiated for the root causality, which is potentially very important given the physical drivers of the hydrology. I think this is a missed opportunity of the authors to expand on more.

- The authors get at this point in their discussion a little (line 132 and 190) but only through examples of two locations. Maybe it’s outside of the scope of this paper to include new subsets of definitions here, but since they are putting forth a new definition, it seems appropriate to at least discuss what these different “flavors” of SSI could/should include.

The metric is intended to be integrative, and this work evaluates the system response and not the underlying components. We acknowledge this comment with textual changes intended to better define “high” vs. “low” SSI by giving a consistent range of values (low as ≤ 0.5 , moderate as $0.5 < SSI \leq 0.75$, and high > 0.75). We have addressed the potential issue of “different flavors” of low-to-moderate-to-high SSI values and also segue into an opportunity for future work with the added sentences:

“Thus, a lower SSI value (0-0.5), even when occurring in mountainous regions that store water as seasonal snow, reveals that the timing and magnitude of water inputs in this area is less strongly dictated by snow water storage than an area with a greater (> 0.75) SSI value, carrying implications for changes in hydrology under current and future climate conditions. Representing the specific source of decreased SSI is beyond the scope of this work, aside from the examples used in Fig 1c, 2b, 4 and S6.” (Lines 438-443)

4. How different are the sine curve definitions of the peak SWI and P compared to the actuals (i.e. the day at which those variables peaked)? In the example show in Figure 1c, for example, the SWI has an outlier peak in April, where it’s visually clear that the seasonal peak occurs closer to June – that’s a good thing. But how representative are these examples of other years, other grid-cell behavior?

We thank the reviewer for these questions about the sine curve’s representation of the raw precipitation and surface water inputs data. The quality of fit of the sine curves to the long-term average data are represented in Fig. S8. While the normalized residual sum-of-squares does not specifically state the exact reason for an imperfect fit (e.g., what is the difference in timing between peak SWI in the raw data versus peak SWI as represented by the sine curve?), this figure does capture the general quality fit of each sine curve (to P and SWI data across both modeled and observed datasets).

The subsets of Figure 1b and c are primarily intended to portray two contrasting examples of the SSI methodology. However, Figure S1 shows the spread of long-term SSI values per pixel, per ecoregion. An analysis of these examples with regards to their representativeness of a larger area or average year is outside the scope of this work. In this way, we have made the purpose of these examples clearer in the manuscript:

“Yet, these contrasting locations, intended to illustrate the SSI concept, behave considerably different with regard to snow water storage, as represented by SSI.” (Lines 123-125)

Line 79 – phrasing the motivation of snowpack and water storage around warming changes alone is too simplistic. What about all the other perturbations and expected changes in extremes and secondary effects (e.g. vegetation shifts, wildfires, etc.)? Consider re-wording here and throughout.

We thank the reviewer for this feedback around a more complete description of climate change and potential/associated responses in the SSI. While additional perturbations and secondary effects were not a focus of this work, we agree that these predicted changes will affect the SSI and are worthy of including and mentioning in this manuscript. We have changed the language around the motivation of this work to “climate change” often instead of solely “warming.” Additionally, we have explicitly mentioned secondary effects in the following pieces of text (changes in **bold**):

*“Complex regional variability in snowpack sensitivity to **climate change** is inter-linked with how water resources are partitioned among evapotranspiration, streamflow, and soil water storage, as protracted snow cover duration aligns water availability with vegetative and atmospheric demand [Barnhart et al., 2016; Foster et al., 2016].”* (Lines 69-72)

*“This perspective on snowfall, snowmelt (magnitude and timing), and seasonal snow water storage targets an unaddressed gap in hydrology by evaluating **how future changes in climate** may modify local and regional water availability through changes in the timing of water inputs to the terrestrial system relative to the timing of precipitation.”* (Lines 82-85)

“Susceptibility to secondary climate perturbations (e.g., extreme events, and changes in vegetation) may also be enhanced in these same areas [Clarke et al., 2022; Sykes, 2009].” (Lines 269-270)

*“Thus, trends in SSI represent a new indicator of the hydrologic sensitivity to **climate change**, signifying the declining ability of western mountains to act as natural water towers, **storing water as snow until later in the year.**”* (Lines 273-275)

- Clarke, B., Otto, F., Stuart-Smith, R., & Harrington, L. (2022). Extreme weather impacts of climate change: an attribution perspective. *Environmental Research: Climate*, 1(1), 012001.
- Sykes, M. T. (2009). Climate change impacts: Vegetation. *eLS*.

Line 116 – Most of the Colorado front range on your map has negative SSI. Similarly, only a very small fraction of the Sierra Nevada on Fig 1a has high SSI. While I like the comparison and the reference here, I think a direct comparison of those regions in their entirety is a stretch.

We thank the reviewer this comment. To clarify, we only evaluated long-term SSI values greater than 0. Thus, while the SSI is lower in the Front Range of Colorado, these SSI values

range primarily from 0-0.5. However, to the reviewer's point of directly comparing the Sierra and the Front Range, we have adjusted the language to emphasize that these two examples do not necessarily represent their entire ecoregion. Those changes are reflected in the manuscript and described/pasted under previous comments. The primary purpose of the Front Range-Sierra Nevada SSI comparison was to portray two contrasting examples for the readership to grasp the concept of the SSI.

Fig 2c-d – Can you provide a possible explanation for the positive changes in SSI at some of these gauges? Furthermore, while line 147 says 80% of SNOTEL show a decrease, it's very hard to see on these graphs. Is the change minimal (yellow colors)? Also, much work on SNOTEL changes has been done by others (see work by Mote and co-authors for example), please consider referencing.

The small aerial increases in SSI were a result of increased precipitation seasonality (as opposed to the decreases in precipitation seasonality that we report in the manuscript). We have qualitatively added this result in the manuscript, at the end of our depiction of Figure 2a-b:

“Small aerial increases in SSI were seen in locations where precipitation seasonality increased due to increases in winter precipitation.” (Lines 153-154)

The decreases in SNOTEL-derived SSI values, shown in Fig 2c, were 0.005/decade (unitless) on average (-0.06/decade maximum). **Significant** decreases in SNOTEL-derived SSI values, shown in Fig 2d, were -0.01/decade on average (-0.06/decade maximum). We have added these values into the manuscript where we report the 80% change and previously when we report declines in VIC-derived SSI (changes in **bold**):

*“Ninety-two percent of the study area with SSI values ≥ 0 exhibited a decrease in SSI (**max decline: -0.03/decade, mean: -0.005/decade**), and the other 8% showed an increase in SSI (Fig. 2a). 25.1% of the study domain exhibited a statistically significant decline in SSI ($p < 0.05$, **max decline: -0.03/decade, mean: -0.01/decade**) while only 0.9% showed a significant increase in SSI (Fig. 2b).”* (Lines 129-133)

*“The maximum decline in SNOTEL-derived SSI was -0.06/decade with an average -0.005/decade. Statistically significant trends in observed SSI ($p < 0.05$) existed for 17.8% of all SNOTEL stations, also largely negative in slope (Fig. 2d; **max decline: -0.06/decade, mean: -0.01/decade**). By ecoregion, the difference in SSI slope between the observed SNOTEL and modeled datasets, ranged from 0.001/decade to 0.083/decade. Declines in SSI in both observed SNOTEL and modeled datasets were consistent with related declines in SNOTEL-reported SWE across this region and time period [Mote et al., 2005; 2018].”* (Lines 157-164)

It's also hard to infer what a 0.1 change versus a 0.2 change in SSI really means. Perhaps if this metric becomes more common, inference about if these differences are/are not substantial would come along with it, but as a reader just introduced to the metric, I find it hard to put into perspective.

We thank the reviewer for this comment. In response, we have added a sentence to acknowledge this point and suggest that more analyses of SSI in the future will build more intuition. In addition, future works could consider percent changes in SSI, as comparing percent changes can be more easily related to relative sensitivity:

“Future work using the SSI will build intuition around SSI changes (unitless) and should consider related percent change.” (Lines 463-465)

Figure S4 – If the Canadian Rockies were excluded, is there still a decline in the western North American SSI decline? The statistically significant cells are few and far between within the CONUS if the northern most regions were neglected, making this almost most applicable to a regional phenomenon.

Yes, there is a significant decline in SSI when averaged across the study area excluding the Canadian Rockies and Columbia Mountains/Northern Rockies (and/or all Canadian grid-cells). This is also the case when averaged spatially and arithmetically and looking at annual data (Fig 2) and decadal data (Fig S4). The relationship is weaker (e.g., $p = 0.043$ when evaluating annual SSI without including the two ecoregions vs. to $p = 0.001$), but remains significant using a p -value threshold of 0.05. In response to this comment, we have added the following:

“The declining trend was also significant when evaluating the modeled SSI while excluding the Canadian Rockies and Columbia Mountains/Northern Rockies ($p < 0.05$), the two ecoregions that expressed the majority of statistically significant SSI declines (not shown).” (Lines 182-185)

Figure 6 – It’s not clear what this graph is plotting. What are the gray lines? Are these a select few anomalously wet or dry years, with the average superimposed with a SSI dot? It’s not evident from the paragraph of text or the caption and needs further clarification.

Across the study area, grid-cell annual average SSI, temperature, precipitation, and precipitation fraction data were evaluated as anomalies (or difference values) from the 64-year mean. The data were then divided into SSI anomaly percentile bins, given the large number of resulting data points ($n = 881,920$ grid-cell years; i.e. 13,780 grid-cells x 64 years). Thus, each point in each panel of Figure 6 is an SSI anomaly percentile bin, reducing the number of points to six (10th, 30th, 50th, 70th, 90th, and 99th percentiles). Temperature, precipitation, and precipitation fraction anomalies were averaged according to the SSI anomaly percentile bins. The gray lines use the same methodology but averaged only across each individual ecoregion. In response to this comment, we have revised the text and figure caption to reflect this more clearly (changes in **bold**):

*“Evaluated across the entire study area, areas and years with greater SSI values (> 0.75) **and thus the largest SSI anomaly percentile bins** were associated with cold and wet conditions in fall and winter months but dry conditions in spring months (Fig. 6a-d, **blue points**). **This result further emphasizes the aforementioned distinction between snow-influenced regions that are dominated by fall and winter snowfall versus spring snowfall.** Anomalously low SSI values (0-0.5) **and thus the smallest SSI anomaly percentile bins** were associated with annually dry conditions (Fig. 6a, **red points**).” (Lines 225-231)*

*“**Averaged by individual ecoregion**, the SSI anomaly percentile bins generally followed the anomalous SSI, precipitation, and temperature patterns seen across the entire study area (Fig. 6a-d, **gray lines**).” (Lines 233-235)*

*“This methodology followed that outlined in Musselman et al. [2021], **where all annual anomalies (SSI, precipitation and temperature)** were calculated as the difference from the respective 64-year mean.” (Lines 494-496)*

Minor

Line 39 - the two sentences starting with “Different from...” could appear earlier in the abstract, perhaps after the SSI is first mentioned to provide a better introduction to the topic.

We agree with the reviewer and hope this change provides a clearer and more efficient introduction to the subject. The abstract now reads:

“Mountain snowpacks act as natural water towers, storing winter precipitation until summer months when downstream water demand is greatest. We introduce a Snow Storage Index (SSI), which represents the annual temporal phase difference between daily precipitation and surface water inputs – sum of rainfall and snowmelt into the terrestrial system – weighted by relative magnitudes. Different from snow water equivalent or snow fraction, the SSI represents the degree to which the snowpack is delaying the timing and magnitude of surface water inputs relative to precipitation, a fundamental component of how snow water storage influences the hydrologic cycle. In western North America, annual SSI has decreased ($p < 0.05$) from 1950-2013 in over 25% of mountainous areas, as a result of substantially earlier snowmelt and rainfall in spring months, with additional declines in winter precipitation. By focusing on the timing and magnitude of snow water storage, the SSI and the trends evaluated herein offer a new perspective on hydrologic sensitivity to climate change which have broad implications for water resources and ecosystems.” (Lines 30-42)

Line 67 - can you define or provide some examples of what you mean by snow water storage dynamics? Do you mean here the amount of water stored in snow? The word “dynamics” makes it sound as though you’re referring to some processes.

We thank the reviewer for this comment. We have deleted the word entirely here and throughout the manuscript.

Fig 2b these insets are really hard to see as they are so small.

We thank the reviewer for this feedback, we have adjusted the inset size of Figure 2b.

Figure 3 – figure quality is very poor, consider rotating to landscape to increase dpi.

We thank the reviewer for this feedback. We have reformatted Figure 3 to increase dpi. Please see this figure pasted above with similar feedback from Reviewer #1.

Reviewer #3:

This paper introduces an interesting approach to quantify the hydrologic significance of the snowpack, the snow storage index (SSI). The SSI is used to conduct a trend analysis in North west America, revealing regions where climate change had a significant impact on the snowmelt contribution to runoff. The paper reads well. The conclusions are supported by a large dataset of ground measurements and the output from a model simulation that was evaluated in a previous study. I have only a few concerns and some suggestions.

We thank the reviewer for both the summary of the work and assessment of its quality. We appreciate the positive remarks and the constructive feedback around more thorough explanations around the derivation of surface water inputs and integration of the SSI metric into literature around additional snow-oriented metrics and flow data (to show a broader applicability of the SSI). We also appreciate the detail-oriented look at the way in which we are averaging SSI, an important step in generating this metric. We have made the respective changes based on the reviewer's feedback below.

In the supplement it is written that "Snowmelt was calculated as the negative change in daily SWE". Why was sublimation neglected?

VIC does consider blowing snow sublimation within its snow model but no lateral transport of wind-blown snow across grid cells [Bowling *et al.*, 2004]. We assumed that latent heat fluxes would be preferentially partitioned to melt rather than sublimation when the snowpack was isothermal [Barnhart *et al.*, 2016; Hood *et al.*, 1999]. These clarifications have been added to the manuscript as follows:

*“Snowmelt was calculated as the negative change in daily SWE. VIC considers blowing snow sublimation within its snow model but no lateral transport of wind-blown snow across grid cells [Bowling *et al.*, 2004]. Our use of ΔSWE follows the approach of Barnhart *et al.*, 2016. In general, with relatively large grid cells, the amount of snow distribution is across grid cells is assumed to be negligible relatively to the snow fluxes within grid cells [Tabler, 2003]. We also assumed that negative changes in SWE, by way of a latent heat flux, were primarily associated with a melt flux as opposed to sublimation based on the assumptions used by previous studies [Barnhart *et al.*, 2016; Hood *et al.*, 1999].”*(Lines 340-347)

- Barnhart, T. B., Molotch, N. P., Livneh, B., Harpold, A. A., Knowles, J. F., & Schneider, D. (2016). Snowmelt rate dictates streamflow. *Geophysical Research Letters*, 43(15), 8006-8016.
- Bowling, L. C., Pomeroy, J. W., & Lettenmaier, D. P. (2004). Parameterization of blowing-snow sublimation in a macroscale hydrology model. *Journal of Hydrometeorology*, 5(5), 745-762.
- Hood, E., Williams, M., & Cline, D. (1999). Sublimation from a seasonal snowpack at a continental, mid-latitude alpine site. *Hydrological Processes*, 13(12-13), 1781-1797.
- Tabler, R.D. (2003), Controlling blowing and drifting snow with snow fences and road design, National Cooperative Highway Research Program Project 20-7(147), Transportation Research Board of the National Academies.

“We averaged modeled and observed SSI as both an arithmetic average (average of grid-cell SSI values) and an average of its constituents (average of area-wide precipitation and surface water inputs)”. Later it is written that “We employed a gridded hydrometeorological dataset that contains spatially and temporally continuous daily meteorological forcings and simulated Variable Infiltration Capacity (VIC) model states and fluxes at 1/16°”. The spatial averaging should account for the fact that the VIC grid is not equal area (otherwise high latitude cells will have to much weight). I could not check if this was actually done since the code to produce the analyses was not made available by the authors.

We thank the reviewer for this comment and feedback. The previous spatial averaging did not account for the difference in VIC grid cells sizes, and thus we have revised the calculation and the figure, providing a revision to our methods below. In response to this comment, the following text has been included:

*“We averaged modeled and observed SNOTEL SSI as both an arithmetic average and an average of its constituents. These spatially averaged values incorporated a weighted grid cell area to account for slight differences in grid cell size at different latitudes [New *et al.*, 2000]. The maximum difference in average annual SSI between weighted and non-weighted spatial averages was 1.5%.”* (Lines 176-180)

*“When averaging SSI values, grid cells were weighted by area, which differed slightly depending on latitude [New *et al.*, 2000].”* (Lines 462-463)

- New, M., Hulme, M., & Jones, P. (2000). Representing twentieth-century space–time climate variability. Part II: Development of 1901–96 monthly grids of terrestrial surface climate. *Journal of climate*, 13(13), 2217–2238.

Methods:

- The area of each cell was calculated as followed:
 - $\text{area}_i = (1/16) * (111.13) * (1/16) * \cos(\text{lat}_i) * 111.13$, where “area_i” is the area of each grid cell and “lat_i” is the latitude of the grid cell in question. 111.13 is the length of a degree latitude.
 - The same results occur when we placed a simple weight to each grid cell, based on its latitude: $\text{weight}_i = \cos(\text{lat}_i * (\pi/180))$, where “weight_i” is the relative weight of the grid cell.
- This aerial weight was applied to daily precipitation and surface water inputs, which were then used in equations 1, 2, and 3 to generate a weighted spatial average SSI.

The spatial average results, including a weighted average versus non-weighted average, were nearly the same. The maximum difference in SSI values between the data sets was 0.03 (unitless) – which would equate to a difference of 1.5% (the full possible range of SSI = (-1,1)). Similarly, the *p*-value and slopes were nearly the same through time (1950–2013): *p* = 0.00337 with weighted averaging compared to *p* = 0.00366 non-weighted averaging, and slope = -0.00405 with weighted averaging compared to slope = -0.00407 non-weighted averaging.

The following figure shows the timeseries of spatially averaged SSI, where the weight/area of each grid cell was considered versus not considered. Error has been temporarily removed from this figure to most clearly depict the slight differences in spatially averaged annual SSI. Trend lines indicate a significant relationship (*p* < 0.05). Fig 3a has been updated accordingly, as well as Fig 3b, which used a spatial average to evaluate decadal changes in SSI. Those data were also near-identical.

We applied the same weighting methodology to the arithmetic average SSI, shown in the following figure, which also revealed near-identical results. Here, the aerial weight was applied to the annual average SSI value per grid cell. The black line is what we showed in Figure 3a in the original version of this manuscript (non-weighted arithmetic average SSI), and the light

blue line is the new weighted arithmetic average SSI data. This has also been updated in Figure 3a.

Surprisingly, the derivation of the SSI formulae is not well explained. The authors fit both P and SWI time series using a sine function but then it is only mentioned that "the two sine curves were combined" but what does "combine" exactly mean?

The explanation of the sine curve methodology has been refined in the Methods section, located at the end of this manuscript, per the journal format. We have updated this section of the methods section prior to equation 3:

“To generate the Snow Storage Index (SSI), the determined phase and amplitude from equations 1 and 2 are mathematically related using a similarity index [Woods et al., 2009].” (Lines 402-404)

- Woods, R.A., (2009), Analytical model of seasonal climate impacts on snow hydrology: Continuous snowpacks, *Adv. Water Resour.*, 32(10), 1465-1481.

Since the paper aims to introduce a new index, I think it would be useful to discuss how it complements previous indices, which also aimed to represent the hydrologic role of snow such as:

- Staudinger et al. (2014), A drought index accounting for snow. *WRR*
- Huning & Aghakouchak (2020), Global snow drought hot spots and characteristics. *PNAS*
- Mankin et al. (2015) The potential for snow to supply human water demand in the present and future. *ERL*
 - In particular, the "snow resource potential" in the Mankin study seems quite similar to the SSI (if yes revise manuscript line 252).

We thank the reviewer for this suggestion. We agree that the SSI would only increase in computational strength if put in the context of other, existing indices. In response to this comment, we added the following paragraph:

“Importantly, the SSI metric quantifies the temporal offset between precipitation and surface water inputs, in addition to the amount of water stored in the snowpack, a combination of hydrologic characteristics not previously captured in a single value. The SSI complements preceding indices aimed at quantifying the integrative hydrologic impact of seasonal snow, such as the temporal lag between precipitation and snowmelt used to identify downstream hydrologic deficits from snowmelt and rainfall [Staudinger et al.,

2014] and the potential for snowmelt to meet downstream demands in hydrologically vulnerable regions of the world [Immerzeel et al., 2020; Mankin et al., 2015].” (Lines 276-283)

L147, "Observed SSI trends across the area (Fig. 2c and d) corroborate the model-based analysis" But looking at fig. 2, model-based and station-based SSI changes do not look really consistent?

We thank the reviewer for this feedback. The direction of change is often regionally consistent. Discrepancies primarily exist in the area above the USA-Canada border. This is because the observed dataset is portraying April 1 SWE, not SSI (due to data availability limitations). Across the entire area, the maximum and average magnitude of change is reported in a previous response to a comment from Reviewer 2 and now in the manuscript. We agree that these two datasets do not show consistent changes in SSI in **all** regions. Yet, on average, each ecoregion shows much consistency, with the largest discrepancies seen in the inter-mountain ecoregions. Between Figure 2a and d, here we report mean slope (change in SSI per year) for [model, observed] across each ecoregion:

- North Cascades/Cascades [-0.0017, -0.0011]
- Sierra Nevada [-0.0010, -0.0022]
- Idaho Batholith [-0.0037, -0.0097]
- Canadian Rockies (south of Canadian border) [-0.0048, -0.0064]
- Southern Rockies [-0.0024, -0.0023]
- Wasatch and Uinta Mountains [-0.009, -0.0160]
- Middle Rockies [-0.0022, -0.0105]

We report select ecoregions, capturing maximum and minimum differences across datasets.

Thus, the range in SSI slopes difference across datasets is 0.0083/year (maximum), and 0.0001/year (minimum). Differences in magnitude across the datasets are expected given the large difference in spatial scale between the VIC grid cells and SNOTEL station data and the differing length of datasets (64 versus 34 years). In response to this comment, we have reported these data in the manuscript as a change per decade:

“Observed SSI trends across the study area corroborate the model-based analysis, with 80% of Snowpack Telemetry (SNOTEL) stations recording a decrease in SSI value over the period of 1984-2018 (Fig. 2c, circular points below the black Canadian border). The maximum decline in SNOTEL-derived SSI was -0.06/decade with an average -0.005/decade. Statistically significant trends in observed SSI ($p < 0.05$) existed in 17.8% of all SNOTEL stations, also largely negative in slope (Fig. 2d; max decline: -0.06/decade, mean: -0.01/decade). By ecoregion, the difference in SSI slope between the observed SNOTEL and modeled datasets ranged from 0.001/decade to 0.083/decade. Declines in SSI in both observed and modeled datasets were consistent with related declines in SNOTEL-reported SWE across this region and time period [Mote et al., 2005; 2018].” (Lines 155-164)

L292, the authors emphasized the uncertainty associated to the rain-snow partition temperature. But why focusing on this aspect? Many other parameterizations could cause model errors. The 6 km resolution of the model is probably a major source of error in high mountain regions, because the model forcing are assumed to be homogeneous over an area with a large meteorological variability (including the air temperature which is used to estimate the rain-snow fraction).

We agree that there are many sources of potential error with regards to the model output. We have refined this statement to suggest that other atmospheric/climatological attributes could affect also surface water input generation, including the resolution of available data:

“Future work should explore alternative rain-snow air temperature thresholds to evaluate associated sensitivities of snowfall fraction and snowmelt model outputs [Jennings et al., 2018; Jennings and Molotch, 2019]. In addition, a more exhaustive exploration of snowmelt model uncertainties (e.g., forest-vegetation) is warranted given their influence on the timing and amount of surface water input generation [Varhola et al., 2010]. Similarly, while the effects of the subsurface are expected to be minimal with regards to surface water inputs, relevant connectivity among soil, groundwater, and vegetation should also be explored, particularly as it pertains to eventual streamflow [Cayan et al., 2010; McNamara et al., 2005]. Finally, higher resolution datasets in complex, mountainous terrain could potentially reduce errors in meteorological forcings [Maina et al., 2020].” (Lines 307-316)

- Varhola, A., Coops, N. C., Weiler, M., & Moore, R. D. (2010). Forest canopy effects on snow accumulation and ablation: An integrative review of empirical results. *Journal of Hydrology*, 392(3-4), 219-233.

Two (optional) suggestions to better show the value of the SSI for future studies:

- High SSI values should be found in mountainous regions under the influence of the Mediterranean climate (cold and wet winter, dry and hot summer, Fayad et al. 2017 JoH). It would be interesting to investigate how the SSI patterns relate to a more standard climate classification (e.g. Koppen).
- It would be interesting to evaluate if the SSI trends spatially match river flow centroid trends across North America (e.g. Stewart et al. 2005 or a more recent study if available).

We thank the reviewer for this suggestion. We agree that a next step in this work is to shed light on the broader application of the SSI in terms of water resources. A supporting manuscript from the first-author’s dissertation is in review. This paper relates this metric to hydrologic partitioning, which builds off of the current statement in this manuscript: *“Hence, further utility in the SSI is expected when evaluating hydrologic partitioning to streamflow and sensitivities to climate change.”* (Line 294-295). We have responded to this suggestion by qualitatively comparing the changes in SSI to the Stewart et al., [2005] paper, particularly to their Figure 4 (d-f). We have added that result/comparison here:

“Because the increase of water availability in March and decrease in July has been documented in flow volumes across the western United States [Stewart et al., 2005], we would expect the SSI and its components to be highly related to the ultimate amount of water which becomes streamflow for downstream users.” (Lines 298301)

We have also referenced the connection of the constituents of the SSI to flow here:

“Concerningly, changes in the volume and timing of water release from mountain snowpacks have cascading effects on streamflow timing and volumes [Stewart et al., 2005] and water storage and conveyance infrastructure, and therefore on regional water availability [Cayan et al., 2001; Harpold et al., 2012; Siirla-Woolburn et al., 2021; Stewart et al., 2004, 2005; Trujillo and Molotch, 2014].” (Lines 56-60)

3rd Feb 23

Dear Dr Hale,

Please allow us to apologise for the delay in sending a decision on your manuscript titled "Recent decreases in snow water storage in western North America". It has now been seen by our reviewers, whose comments appear below. In light of their advice I am delighted to say that we are happy, in principle, to publish a suitably revised version in Communications Earth & Environment under the open access CC BY license (Creative Commons Attribution v4.0 International License).

We therefore invite you to revise your paper one last time to address the remaining concerns of our reviewers. At the same time we ask that you edit your manuscript to comply with our format requirements and to maximise the accessibility and therefore the impact of your work.

EDITORIAL REQUESTS:

*****Please take care to match our formatting and policy requirements. We will check revised manuscript and return manuscripts that do not comply. Such requests will lead to delays. *****

SUBMISSION INFORMATION:

OPEN ACCESS:

Communications Earth & Environment is a fully open access journal. Articles are made freely accessible on publication under a [CC BY license](http://creativecommons.org/licenses/by/4.0) (Creative Commons Attribution 4.0 International License). This license allows maximum dissemination and re-use of open access materials and is preferred by many research funding bodies.

For further information about article processing charges, open access funding, and advice and support from Nature Research, please visit <https://www.nature.com/commsenv/article-processing-charges>

At acceptance, you will be provided with instructions for completing this CC BY license on behalf of

all authors. This grants us the necessary permissions to publish your paper. Additionally, you will be asked to declare that all required third party permissions have been obtained, and to provide billing information in order to pay the article-processing charge (APC).

[link redacted]

Best regards,

Christophe Kinnard, PhD
Editorial Board Member
Communications Earth & Environment
orcid.org/0000-0002-4553-5258

Joe Aslin
Senior Editor,
Communications Earth & Environment
<https://www.nature.com/commsenv/>
Twitter: @CommsEarth

REVIEWERS' COMMENTS:

Reviewer #1 (Remarks to the Author):

In my opinion, authors considerably improved the manuscript and they satisfactorily responded to my comments. I appreciate that authors added some more information and discussion about the using of the new metric. All changes made the manuscript more consistent and convincing. Therefore, I can recommend the manuscript for publication. I have only two following technical comments.

- Reference "Staudinger et al., 2014" is not included in the reference list. Besides, I think authors means paper by Staudinger et al. 2014 (without "n"); probably the one published in WRR.
- Please doublecheck that Fig. S6 is correctly uploaded to the editorial system since it didn't appear both in rebuttal letter and pdf version of the revised manuscript (docx version seems to be correct).

Michal Jenicek (Reviewer #1)

Reviewer #3 (Remarks to the Author):

The authors have adequately addressed my main concerns but there remain three points to clarify.

1) Sublimation. It is now acknowledged in the manuscript that the snowpack ablation (negative SWE change) was assumed to be equal to the melt, which means that SWE losses by sublimation were neglected. I wonder why the authors made this assumption since the VIC model computes the latent heat flux. Can the author clarify this point or provide an estimation of the range of % SWE loss by sublimation in their domain to justify this simplification?

2) Spatial averaging method. This point was well addressed by the authors and the updated results do not modify their conclusions. However, I see no reason to keep the mentions of the previous "arithmetic averages" in the manuscript since these calculations were not correct.

3) SSI Equation. I still don't understand Eq 3:

$$\delta_{\text{SWI}} \cdot \text{sgn} \cdot \delta_{\text{P}}$$

if the sign function is applied to δ_{P} (?) then the dot should be removed. In general, the equations would be clearer and more conform to standards by removing the bold mid-dots. The "minus one" in the end of Eq 3 could also be replaced by a minus sign at beginning of the equation. Finally, sin and cos functions should not be italicized (use the same format as the sgn function).

Response to reviewers:

We thank the reviewers for their added time and insightful comments and suggestions regarding the updated version of our manuscript. We have addressed each point below using green, indented text. References to section titles correspond to the final manuscript draft. In summary, we better organized our final figures, reference list, and equations, and we clarified uncertainty around sublimation within our methods section.

Below are our general and section-specific responses to the two reviewers' comments.

Reviewer #1:

In my opinion, authors considerably improved the manuscript and they satisfactorily responded to my comments. I appreciate that authors added some more information and discussion about the using of the new metric. All changes made the manuscript more consistent and convincing. Therefore, I can recommend the manuscript for publication. I have only two following technical comments.

We thank the reviewer for this overall assessment, and we are glad that the reviewer sees the contribution of this targeted work to broader water resources readership and scientific communities.

- Reference “Staudinger et al., 2014” is not included in the reference list. Besides, I think authors means paper by Staudinger et al. 2014 (without “n”); probably the one published in WRR.
- We have revised this in-text citation to read “Staudinger et al., 2014” (removing the misspelled “Staundinger”) and placed the appropriate full citation in our reference list.
- Please doublecheck that Fig. S6 is correctly uploaded to the editorial system since it didn't appear both in rebuttal letter and pdf version of the revised manuscript (docx version seems to be correct).
- We have included the appropriate Figure S6 within our Supplemental Material file.

Reviewer #3:

The authors have adequately addressed my main concerns but there remain three points to clarify.

- 1) Sublimation. It is now acknowledged in the manuscript that the snowpack ablation (negative SWE change) was assumed to be equal to the melt, which means that SWE losses by sublimation were neglected. I wonder why the authors made this assumption since the VIC model computes the latent heat flux. Can the author clarify this point or provide an estimation of the range of % SWE loss by sublimation in their domain to justify this simplification?

We agree with the reviewer that it is important to clearly state the uncertainty around sublimation and also clarify our rationale for not including this variable in our SSI calculation. As such, we have revised the following sentences in section “Datasets:”

“We assumed that negative changes in SWE, by way of a latent heat flux, were primarily associated with a melt flux as opposed to sublimation, which may introduce some uncertainty in our analysis [Barnhart et al., 2016; Etchevers et al., 2004; Hood et al., 1999]. However, we would not expect small residuals of snow lost to sublimation to affect our analysis, since the SSI trends were consistent across differences in climate and relative humidity, known drivers of snowpack ablation [Harpold and Brooks, 2018].”

Added citations:

- Etchevers, P., Martin, E., Brown, R., Fierz, C., Lejeune, Y., Bazile, E., ... & Yang, Z. L. (2004). Validation of the energy budget of an alpine snowpack simulated by several snow models (Snow MIP project). *Annals of Glaciology*, 38, 150-158.
- Harpold, A., Brooks, P., Rajagopal, S., Heidebuchel, I., Jardine, A., and Stielstra, C., (2012), Changes in snowpack accumulation and ablation in the intermountain west, *Water Resources Research*, 48(11), W11501, doi:10.1029/2012WR011949.

- 2) Spatial averaging method. This point was well addressed by the authors and the updated results do not modify their conclusions. However, I see no reason to keep the mentions of the previous "arithmetic averages" in the manuscript since these calculations were not correct.

Thank you for this comment, we have removed the arithmetic spatial averaging results from our analysis, reporting only the spatially-weighted results.

- 3) SSI Equation. I still don't understand Eq 3: $\delta_{SWI} \cdot \text{sgn} \cdot \delta_P$. If the sign function is applied to δ_P (?) then the dot should be removed. In general, the equations would be clearer and more conform to standards by removing the bold mid-dots. The "minus one" in the end of Eq 3 could also be replaced by a minus sign at beginning of the equation. Finally, sin and cos functions should not be italicized (use the same format as the sgn function).

Thank you for pointing out these errors. Equations 1-3 have been reformatted to match standard equation practices and those outlined by the journal. Equation 3, in particular, now reads: $SSI = -[\delta_{SWI} \text{sgn}(\delta_P) \cos\left(\frac{2\pi(s_{SWI}-s_P)}{365}\right)]$ (3)